# Single cell RNA-seq reveals profound transcriptional similarity between Barrett's oesophagus and oesophageal submucosal glands

Richard Peter Owen [1], Michael Joseph White[1], David Tyler Severson [1], Barbara Braden [2], Adam Bailey[2], Robert Goldin[3], Lai Mun Wang[4], Carlos Ruiz-Puig[1], Nicholas David Maynard[5], Angie Green[6], Paolo Piazza[6,9], David Buck[6], Mark Ross Middleton [7], Chris Paul Ponting [8], Benjamin Schuster-Böckler [1] & Xin Lu [1]

Barrett's oesophagus is a precursor of oesophageal adenocarcinoma. In this common condition, squamous epithelium in the oesophagus is replaced by columnar epithelium in response to acid reflux. Barrett's oesophagus is highly heterogeneous and its relationships to normal tissues are unclear. Here we investigate the cellular complexity of Barrett's oesophagus and the upper gastrointestinal tract using RNA-sequencing of single cells from multiple biopsies from six patients with Barrett's oesophagus and two patients without oesophageal pathology. We find that cell populations in Barrett's oesophagus, marked by *LEFTY1* and *OLFM4*, exhibit a profound transcriptional overlap with oesophageal submucosal gland cells, but not with gastric or duodenal cells. Additionally, SPINK4 and ITLN1 mark cells that precede morphologically identifiable goblet cells in colon and Barrett's oesophagus, potentially aiding the identification of metaplasia. Our findings reveal striking transcriptional relationships between normal tissue populations and cells in a premalignant condition, with implications for clinical practice.

---

[1] Ludwig Institute for Cancer Research, Nuffield Department of Medicine, University of Oxford, Oxford OX3 7DQ, UK. [2] Translational Gastroenterology Unit, Nuffield Department of Medicine, University of Oxford, Oxford OX3 9DU, UK. [3] Centre for Pathology, St Mary's Hospital, Imperial College, London W2 1NY, UK. [4] Department of Pathology, Oxford University Hospitals NHS Foundation Trust, Oxford OX3 9DU, UK. [5] Department of Upper GI Surgery, Oxford University Hospitals, Oxford OX3 7LE, UK. [6] Wellcome Trust Centre for Human Genetics, University of Oxford, Oxford OX3 7BN, UK. [7] Department of Oncology, Old Road Campus Research Building, Roosevelt Drive, Oxford OX3 7DQ, UK. [8] MRC Human Genetics Unit, MRC IGMM, University of Edinburgh, Crewe Road, Edinburgh EH4 2XU, UK. [9] Present address: Department of Medicine, Faculty of Medicine, Imperial College London, London W12 0NN, UK. These authors contributed equally: David Tyler Severson, Richard Peter Owen, Michael Joseph White. Correspondence and requests for materials should be addressed to B.S.-B. (email: benjamin.schuster-boeckler@ludwig.ox.ac.uk) or to X.L. (email: xin.lu@ludwig.ox.ac.uk)

At least 80% of cancers arise from epithelial cells. In many tumours a change in cell type, referred to as metaplasia, is a key step in cancer initiation. Barrett's oesophagus (BO) is an example of metaplasia in the distal oesophagus and affects 1 in 50 people[1]. BO is defined as replacement of squamous epithelium by columnar epithelium, and it gives a 30-fold increased risk of developing oesophageal adenocarcinoma (OAC) which has a five year survival of only 15%[2–4]. BO is associated with gastro-oesophageal reflux disease, suggesting it occurs in response to a chronically inflamed environment[5]. Remarkably, several anatomically distant cell types are also identifiable in BO, most commonly intestinal goblet cells but also Paneth and pancreatic acinar cells, among others[6–8].

This apparent plasticity in BO has obscured its relationship with normal gastrointestinal (GI) tissues, as no normal GI tissue is as heterogeneous as BO. Several theories are proposed for the origin of BO. A widely held view is that BO originates from the stomach[9,10], and studies looking for similarities (e.g. in gene or protein expression and cellular appearance) between BO and selected normal tissues - including the intestine, gastric pylorus, gastric corpus and gastric cardia – have found some shared attributes[11,12]. There is also evidence suggesting BO may originate directly from native oesophageal squamous[13] or submucosal gland cells[14–17], from recruitment of circulating stem cells[18], or from reactivation of dormant p63[−]/KRT7[+] residual embryonic cells (RECs) in situ[19]. In contrast to p63[−]/KRT7[+] RECs, a recent study identified p63[+]/KRT5[+]/KRT7[+] cells derived from the squamocolumnar junction as the cells of origin of BO in a transgenic mouse model with ectopic expression of CDX2 in KRT5[+] epithelium[20]. Many of the proposed BO origin theories are based on transgenic mouse studies, and the submucosal gland cell theories are based on human histopathology studies. Unfortunately, submucosal gland theories cannot be tested in mice since mice and humans have key differences in their gastrointestinal anatomy, and rodents lack oesophageal glands[21]. These difficulties argue for an unbiased and systematic genetic approach to BO characterisation in humans with all relevant control cell types to better understand the origin of BO cell types.

Single cell RNA-sequencing (RNA-seq) combined with computational methods for functional clustering of cell types provides a less biased approach to understanding cellular heterogeneity. Given the highly heterogeneous nature of BO, we hypothesise that single cell RNA-seq might clarify the relationships between cells in normal tissues and BO, and indicate whether there are specialised cells in BO with similar functions to cells elsewhere in the gastrointestinal tract. Therefore we apply this approach to biopsies from BO, normal oesophagus, stomach and small intestine (duodenum). This reveals a cell population in BO that expresses the developmental gene (LEFTY1) and is distinct from intestinal or gastric cells, but has a highly similar RNA composition to columnar gene expressing cells from oesophageal submucosal glands in normal oesophagus.

## Results

### Single cell RNA-seq identifies subpopulations in normal upper GI epithelia

To characterise the cell populations in BO, samples were taken from 13 BO patients (A-D, I-Q) attending for routine endoscopic surveillance of non-dysplastic BO. From each patient, we took biopsies from BO, adjacent macroscopically normal oesophagus (20 mm proximal to BO), stomach (20 mm distal to the gastro-oesophageal junction) and duodenum (Fig. 1a). Individual 2 mm biopsies were divided to provide tissue for single cell RNA-seq, bulk tissue RNA-seq and histology in 4 out of 13 patients, and bulk tissue RNA-seq and histology alone in the remaining 9 patients (see Methods). Single cells and histology

were also prepared from normal oesophageal biopsies from two patients with gastro-oesophageal reflux disease but no previous or current diagnosis of BO or any other oesophageal pathology. All sampled patients were taking regular acid suppression therapy and had no features of oesophageal dysplasia or malignancy (Supplementary Table 1).

Bulk RNA-sequencing followed by hierarchical clustering of differentially expressed genes in the duodenal, gastric, oesophageal and BO samples from 13 patients with BO showed a clear distinction between squamous (i.e. normal oesophagus) and non-squamous (i.e., gastric, duodenum and BO) epithelia (Fig. 1b). BO samples from all 13 patients had some similarities to duodenal and gastric samples (Fig. 1b). When a defined list of genes known to distinguish gastrointestinal epithelia[12] was used in hierarchical clustering, BO samples appeared most closely related to gastric tissue, consistent with previous studies[22] (Fig. 1c).

For single cell RNA-seq, a total of 4237 cells were sequenced from 8 patients (Supplementary Table 1) in three batches. Due to known issues with batch effects in single cell experiments[23], analysis of cells from each batch has been kept separate where feasible and cells were permuted across plates and pooled prior to sequencing (see Methods). The first batch yielded 1040 cells (207 duodenum, 227 gastric, 371 BO and 235 oesophagus) suitable for analysis from four patients (A-D) with BO and intestinal metaplasia. A total of 214, 35, 66 and 56 BO cells were analysed from each BO patient, respectively. The second batch yielded 648 oesophagus cells suitable for analysis from two patients (E-F) with symptoms of gastro-oesophageal reflux but no identifiable oesophageal pathology. Finally, the third batch of cells yielded 194 cells (29 pylorus, 109 gastric, 32 BO and 24 oesophagus) suitable for analysis from two patients (G-H) with BO and intestinal metaplasia. Overall, there was a mean of $1.2 \times 10^5$ gene counts per cell and a median of 3978 genes were detected per cell (with at least one count per gene).

First, we clustered the cells from each normal tissue type from the BO patients by gene expression (Fig. 1d). The eleven clusters (D1-D4, G1-G3 and O1-O4, in duodenum, gastric and oesophageal samples, respectively) were then annotated on the basis of genes previously characterised as expressed in specific cell types (complete list in Supplementary Data 1). In the duodenum, these are: intestinal alkaline phosphatase (ALPI)-expressing enterocytes (D1); mucin 2 (MUC2)-expressing goblet cells (D2); olfactomedin 4 (OLFM4)-expressing crypt cells (D3); and some uncharacterised cells expressing Joining Chain Of Multimeric IgA And IgM (JCHAIN) (D4). In the gastric samples, these are: chromogranin (CHGA)-expressing enteroendocrine cells (G1); gastrokinin (GKN1)- and trefoil factor 1 (TFF1)-expressing foveolar cells (G2); and mucin 6 (MUC6)- and TFF1-expressing mucus neck cells (G3). Of note, the proton pump gene ATP4A and the intrinsic factor gene GIF were rarely detectable in gastric cells, indicating these are cardiac-type gastric samples (Supplementary Fig. 1).

Interestingly, four clusters were identified in the oesophageal samples. Two of these express expected squamous genes (KRT5, KRT14, TP63; clusters O1 and O2) and two express the columnar gene TFF3 (clusters O3 and O4). The two squamous clusters can be distinguished by the presence (O1) or absence (O2) of acute phase response (SAA1) gene expression, presumably representing squamous cells in different states. The detection of TFF3 in O3 and O4 is of great interest and is consistent with these cells being from the columnar epithelium of oesophageal submucosal glands (OSGs)[24], a structure in the normal human oesophagus. To validate this, we used samples of normal oesophagus taken from the proximal part of an oesophagectomy specimen following resection for a Siewert type III junctional tumour to illustrate the structure of OSGs, OSG ducts and squamous epithelium (Fig. 1e).

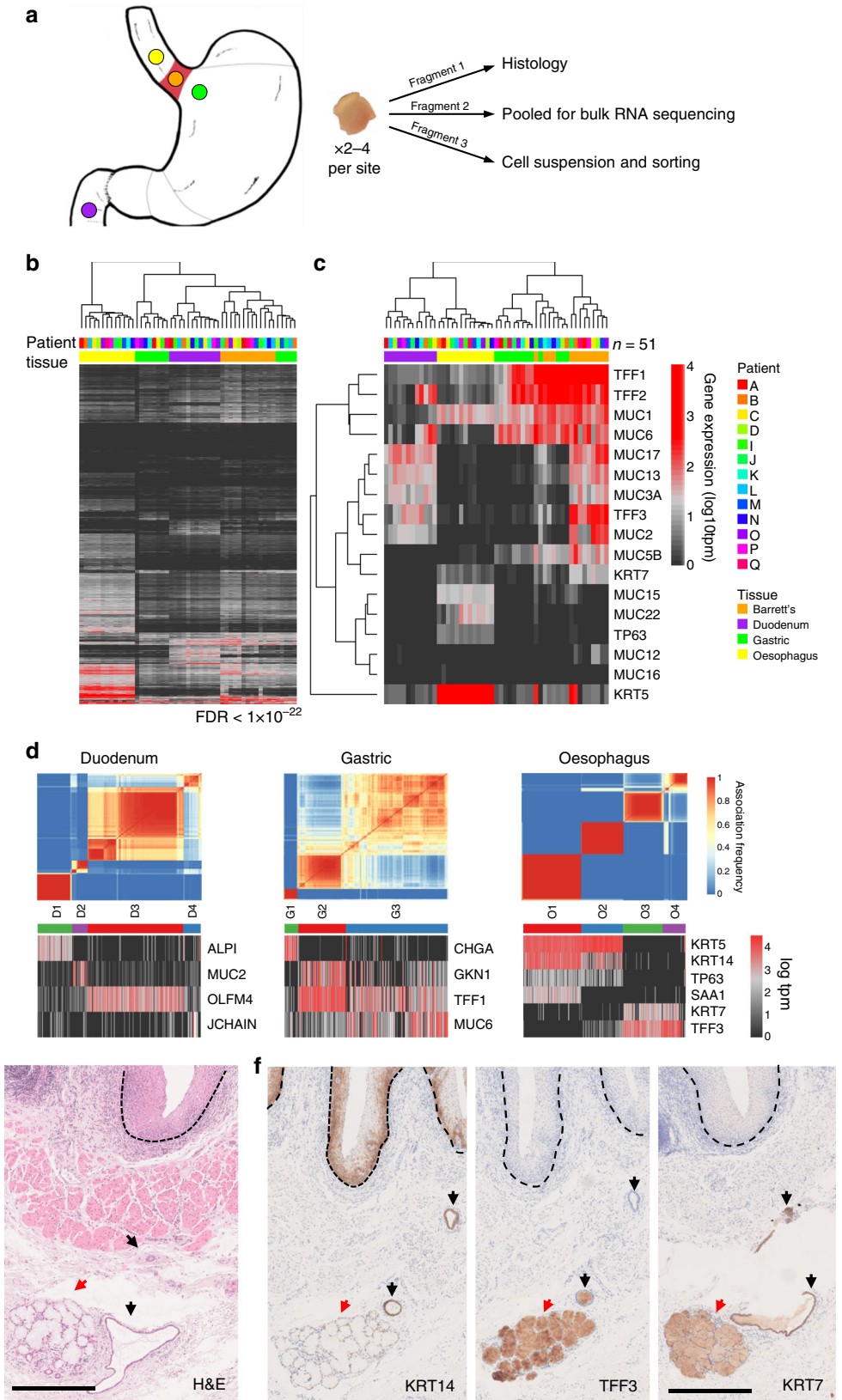

Since OSGs comprise different cell lineages, including squamous lineages, we detected cytokeratin 14 (KRT14, a squamous cell marker)-expressing cells in OSG ducts, demonstrating they are bona fide OSGs. Using the adjacent sections from the same OSG-containing specimen, we observed TFF3 and keratin 7 expression in OSG structures exclusively (Fig. 1f). These results show that single cell transcriptomic analysis can identify gastrointestinal epithelial cell subpopulations, including cells from OSGs that cannot be distinguished by conventional bulk RNA-seq.

**Fig. 1** Single cell RNA sequencing identifies cell groups in normal upper gastrointestinal epithelia. **a** Endoscopic sampling sites (yellow, oesophagus; green, gastric cardia; purple, duodenum; orange, Barrett's oesophagus) with summary of how tissues from patients were used. Two to four biopsies were taken at each site. Patients without BO were sampled from the lower oesophagus 20 mm proximal to the squamous-columnar junction. **b** From bulk RNA-seq data derived from samples from 13 patients with BO, heatmap of genes differentially expressed between any tissue type (analysis of variance-like test, false discovery rate (FDR) $<1 \times 10^{-22}$) with tissue hierarchy determined by nearest neighbour. Tissue indicated by colours as in **a**. One duodenal sample from patient Q failed to produce usable data and was excluded. **c** From bulk RNA-seq data, heatmap of expression of mucin and trefoil factor genes with tissue hierarchy determined by nearest neighbour, in samples from 13 patients with BO. **d** Upper panels show the cluster consensus matrices for single cells from normal tissue sites in four BO patients. Blue-to-red colours denote the frequency with which cells are grouped together in 250 repeat clusterings of simulated technical replicates (see Methods). Cell clusters are indicated by coloured bars below the matrices. In lower panels, heatmaps show expression of known functionally relevant genes that were differentially expressed between cell clusters (>4 fold change, FDR $<1 \times 10^{-5}$). **e** Haematoxylin and eosin staining of normal oesophagus taken from the proximal part of an oesophagectomy specimen resected for Siewert type III junctional tumour in a patient with no BO, showing OSGs (red arrow), OSG ducts (black arrow), and squamous epithelium (marked with dotted black line). Scale bar, 500 μm. **f** Immunohistochemical staining of KRT14, TFF3 and KRT7 (left, middle and right images, respectively) in adjacent sections from the same specimen as **e**, showing OSG ducts (black arrows) and OSGs (red arrows) and squamous epithelium (marked with dotted black line). Scale bar, 500 μm. OSG oesophageal submucosal gland

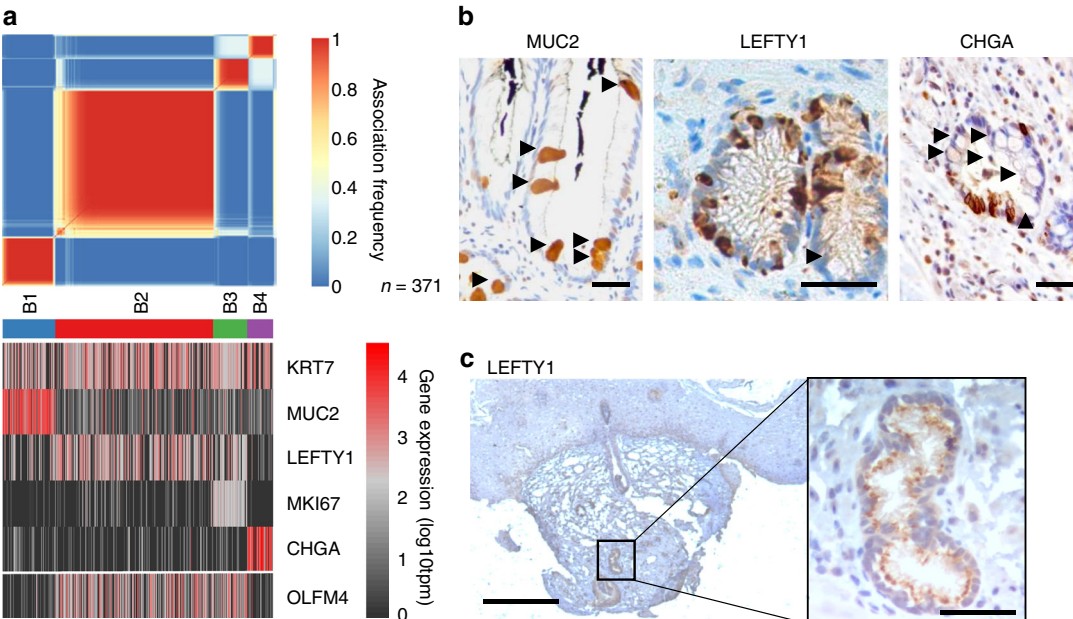

**Fig. 2** *LEFTY1* and *OLFM4* are mainly expressed in Barrett's oesophagus cells that do not express differentiated secretory cell markers. **a** Upper panel, cluster consensus matrix of BO cells from 4 BO patients ($n = 371$ cells). Blue-to-red colours denote the frequency with which cells are grouped together in 250 repeat clusterings of simulated technical replicates (see Methods). Clusters (B1-B4) are indicated by the coloured bars below. Lower panel, heatmaps showing expression of selected functionally relevant genes that are differentially expressed between cell clusters (>4 fold change, FDR <1e-5).
**b** Immunohistochemical staining of MUC2, LEFTY1 and CHGA in sections derived from the same BO resection specimen. Black arrows indicate goblet cells on all sections (positively stained for MUC2; negative for LEFTY1 and CHGA). Scale bars are 50 μm. **c** Immunohistochemical staining of LEFTY1 in an OSG from a normal squamous endoscopic biopsy obtained from a patient with BO. Scale bars are 300 μm and 50 μm in enlarged image

**Barrett's oesophagus is enriched for *LEFTY1*-expressing cells.** To identify genes characteristic of distinct BO cell populations we clustered all the BO cells by gene expression (Fig. 2a, also see Supplementary Data 1). The clusters (B1-B4) can be distinguished by expression of *MUC2* (B1; goblet cells, 19% of BO cells); *LEFTY1* (B2 and B3, 71% of BO cells); and *CHGA* (B4; enteroendocrine cells, 9.7% of BO cells). Since all patients had intestinal metaplasia, goblet cells made up 22%, 2.9%, 29% and 7.1% of cells in patient A-D, respectively. *KRT7* is expressed similarly across all 4 clusters, consistent with it being a marker of BO[25,26]. The *LEFTY1*-expressing cells (B2 and B3; Fig. 2a) are divided into a larger, low proliferating (*MKI67* (Ki67) negative) cluster (B2) and a smaller, high proliferating (*MKI67* positive) cluster (B3). LEFTY1, a secreted protein and transforming growth factor beta (TGF-β) superfamily member, is normally expressed

in development, where it has roles in left-right asymmetry determination[27], but little is known about its potential roles in adult tissues and it has not previously been associated with BO.

To confirm the above finding and to further characterise LEFTY1 expression, we first examined MUC2, LEFTY1 and CHGA expression in sections generated from the same BO resection specimen. LEFTY1 expression was detected in BO epithelial cells (Supplementary Data 2). Interestingly, morphologically identifiable goblet cells are positive for MUC2 but not LEFTY1 or CHGA (Fig. 2b).

To further characterise LEFTY1 expression, we stained 140 BO samples from 80 patients, 78 endoscopic biopsies from control sites (oesophagus, gastric fundus and duodenum) in 26 BO patients, and additionally five endoscopic samples from the pylorus, five resected samples of normal colon and five samples of

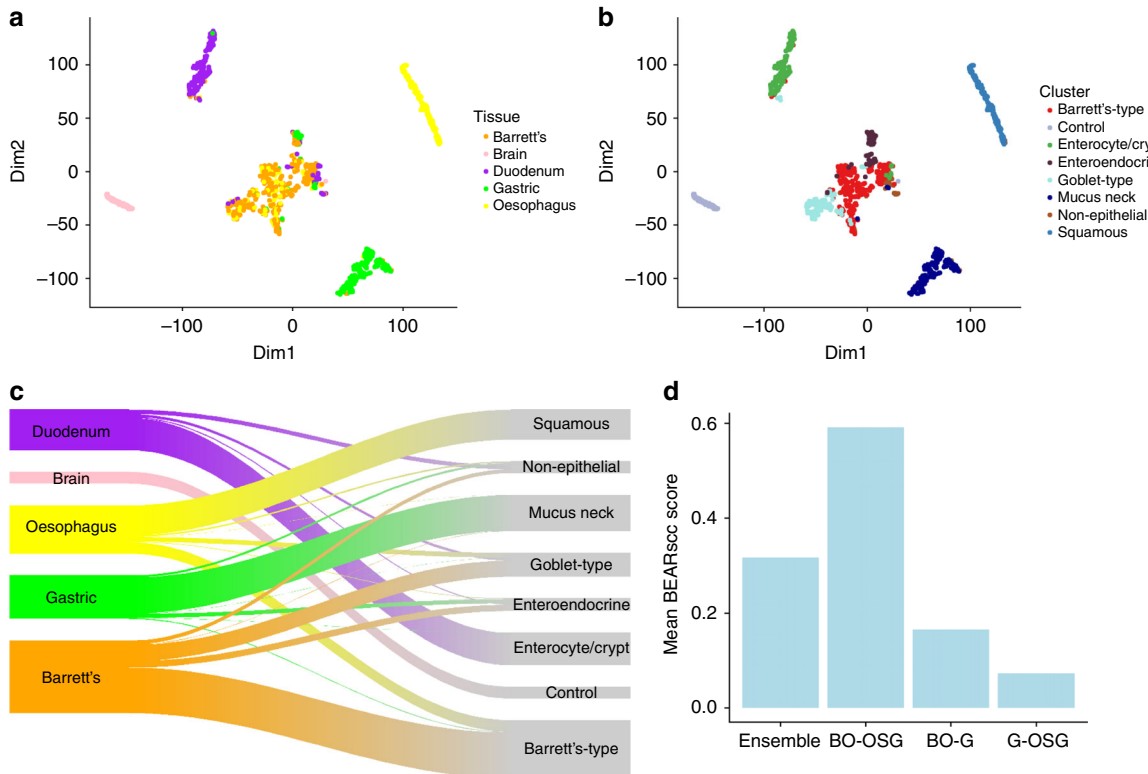

**Fig. 3** The majority of Barrett's oesophagus cells have a similar transcript profile to oesophageal submucosal gland (OSG) cells. **a** t-Distributed Stochastic Neighbour Embedding (t-SNE) plots of cells from all samples from four BO patients ($n = 1107$ including brain control), showing similarity of cells in two dimensions, coloured by tissue type (yellow, oesophagus; green, gastric cardia; purple, duodenum; orange, Barrett's oesophagus; pink, brain). Brain was used as a control. **b** t-SNE plot of cells from four BO patient samples (**A–D**), as in **a**, coloured by how cells contribute to clusters generated by SC3 analysis with 250 repeat clusterings of simulated technical replicates (see Methods). Names given to the clusters are based on expression of known marker genes (see text and Supplementary Fig. 3). **c** Sankey diagram showing how each tissue type sampled contributes to the clusters shown in **b**. Colours and labels on the left indicate sampled tissue (as in **a**); colours and labels on the right indicate cluster (as in **b**). **d** Mean BEARscc score for each grouping of 'gland-like' cells ($n = 372$), which are a sub-set of gastric (G, $n = 175$), BO ($n = 78$) and OSG cells ($n = 119$): excluding gastric and BO cells that expressed *CHGA* or *MUC2* (to exclude enteroendocrine and goblet cells, respectively) and excluding oesophageal cells that did not express *TFF3* (to exclude squamous cells). 'Ensemble' refers to all cells grouped together. Thresholds were set at the tenth centile of cells in which at least one transcript was detected from each gene of interest

normal oesophagus taken from the proximal part of an oesophagectomy specimen resected for junctional tumours (Supplementary Data 2). Overall there are two different *LEFTY1* staining patterns: intensely positive cytoplasmic staining and moderate cytoplasmic staining. Moderate *LEFTY1* staining only, was seen in the Brunner's gland of the duodenum and in the lower portion of the glands in the gastric fundus. In the colon there are a few, intensely positively *LEFTY1* staining cells. Both moderate and intensely expressing *LEFTY1* cells are present in the gastric pylorus and BO (Supplementary Fig. 2). Immunohistochemical staining of oesophageal samples showed that the squamous epithelium was negative for *LEFTY1* staining, as were the OSGs in oesophagectomy samples from non-BO patients. All three OSGs from the 140 oesophageal samples showed moderate cytoplasmic staining throughout the OSG (Fig. 2c). These expression patterns explain why the more superficial mucosal biopsies obtained for single cell RNA-seq show dramatic differences in *LEFTY1* expression between tissues.

**OSGs share an RNA composition profile with Barrett's oesophagus.** Taking all cells from BO patients together (A–D), the normal tissue cells separate clearly from the BO cells based on their gene expression, with the exception of specialised cell types such as goblet or enteroendocrine cells, but the majority of BO cells overlap with a sub-set of oesophageal cells, as seen in a t-Distributed Stochastic Neighbor Embedding (t-SNE) plot (Fig. 3a). Clustering by gene expression (by the same method as in Fig. 1d) assigned cells to 7 clusters (with brain controls in a separate cluster) (Fig. 3b, c, also see Supplementary Fig. 3a). Most of these clusters are similar to those identified in the analysis of normal tissue alone (Fig. 1d) and they can be related to known cell types based on expression of previously characterised genes (Supplementary Fig. 3b, also see Supplementary Data 3 for complete list). The majority of duodenal cells fall in the cluster categorised as 'enterocytes' (similar to D1), gastric as 'mucus neck' (similar to G3) and a substantial proportion of oesophageal cells are in the 'squamous' cluster (similar to O1/O2) (Fig. 3c). Some oesophageal cells, BO cells and a few duodenal cells fall into a 'goblet' cluster, and some gastric cells cluster with a few BO cells in the 'enteroendocrine' cluster. The group described as 'non-epithelial' contains some endothelial cells and *CD45*-low immune cells (Supplementary Fig. 4). Notably, the majority of BO cells (63%) are in the cluster labelled as 'Barrett's-type' that also contains the subset of oesophageal cells that have a gene expression profile consistent with their being OSGs (Fig. 3c, also see Supplementary Data 3). These cells are enriched for *LEFTY1* expression.

To test whether this relationship between BO and native oesophageal cells with columnar characterisation was also seen in patients without BO, we clustered all normal oesophageal cells

from patients with and without BO (A, B, D and E, F, respectively). This showed that cells grouped into five clusters (Supplementary Fig. 5a), three clusters (1, 2 and 4) were mainly squamous and the remaining two (3 and 5) had more columnar marker-expressing cells. Of the 'columnar' clusters, cluster 5 consisted of cells from patients A and B and cluster 3 consisted of cells from patients B, D and E (patients A, B, D had BO, patients E, F had no BO) (Supplementary Fig. 5b). Although rare in these data, it is interesting that one of the clusters (cluster 3) containing *TFF3*[+] cells also had four cells which were positive for the squamous genes *KRT14* (a gene pair with *KRT5*), *TP63* and *KRT7* (Supplementary Fig. 5c). As p63[+] KRT7[+] cells have been shown to generate intestinal-like epithelial cells in organoid culture upon CDX2 overexpression, it may be possible that these oesophageal cells could be related to the transitional zone progenitor cells previously observed in humans[21].

To confirm whether the relationship between BO cells and OSGs was stronger than the associations with other gland-type cells, we looked across the RNA compositions of cells from other tissues, i.e. gastric gland cells and BO cells that did not express *CHGA* or *MUC2* (to exclude enteroendocrine and goblet cells, respectively; see Methods for thresholding), and oesophageal cells that expressed *TFF3* (to exclude squamous cells, Supplementary Fig. 5d-e). We also developed BEARscc, an algorithm which uses external controls to simulate technical replicates to check whether a single cell clustering method is robust to technical variability[28]. The 'score' metric of BEARscc reflects how frequently cells within a group cluster together, as opposed to with cells from other clusters. We compared manually selected groups of (1) gastric and BO cells, (2) gastric and OSG cells and (3) BO and OSG cells, from patients with BO (A-D). The BO and OSG cell combination had a higher score than any combination which included gastric cells, or all cells grouped together, suggesting BO and OSG cells have the most stable cell type relationship (Fig. 3d). Using only these manually selected gastric, BO and OSG cells with additional OSG cells from patients without BO (E-F), unbiased clustering with SC3 also confirmed the strong relationship between BO and OSG cells, with only very few gastric cells clustering with BO or OSG cells (Supplementary Fig. 6a). t-SNE, with the inclusion of duodenal cells which expressed the highest levels of *MUC6* to enrich for duodenal Brunner's gland-type cells (Supplementary Fig. 6b), also confirmed the strong relationship between BO and OSG cells (Supplementary Fig. 6c). This relationship was characterised by high *LEFTY1* expression (Supplementary Fig. 6d). Only a small number of genes show differential expression between BO cells and OSG cells that did not express *CHGA* and *MUC2* (to exclude enteroendocrine and goblet cells). Pathway analysis on these genes did not suggest any biological processes that mechanistically distinguish BO and OSG cells (Supplementary Fig. 6e-f).

In view of the phenotypic overlap with BO and gastric pylorus, we analysed the transcriptomes of 194 cells from an additional two patients (G-H) with BO (24 oesophageal cells, 32 BO cells, 109 gastric cardia cells and 29 gastric pyloric cells). Clustering of these cells on global and specific gene expression show that gastric cardia and pylorus exhibited similar RNA composition properties (Supplementary Fig. 7). The BO cells also expressed several of the gastric genes, but showed differences such as increased *KRT7* and *BPIFB1* expression (Supplementary Fig. 7b). Collectively, these data show that oesophageal cells expressing genes seen in OSGs, and not intestinal, gastric or squamous cells, have the greatest RNA composition similarity to BO cells.

**ITLN1 and SPINK4 mark early goblet cells**. In this study, 19% of BO cells were classified as 'goblet' cells, which is consistent with

the requirement in some countries, such as the US[29], for goblet cells to be present for the diagnosis of BO. Goblet cells are classically defined by morphological appearance and MUC2 expression. Applying a threshold set at the tenth centile to include 90% of cells in which at least one transcript was detected from each gene of interest (to reduce biological noise), we found that *MUC2* RNA co-expressed with intelectin 1 (*ITLN1*) and Kazal type 4 serine peptidase inhibitor (*SPINK4*) in 61% of goblet cells from duodenum, gastric and BO samples (Fig. 4a–b). ITLN1 and SPINK4 have been previously shown to mark goblet cells in the normal gut and some non-gastrointestinal tissues[30,31], but we observed some cells in each tissue type that uniquely expressed *MUC2*, *ITLN1* or *SPINK4*. Therefore we hypothesised that their expression pattern might mark stages of goblet cell development in vivo. To test this, we analysed expression of these proteins by immunofluorescence staining of five human colon samples (approximately 500 crypts examined in each sample). ITLN1 and SPINK4 co-staining was consistently present near the crypt base, where undifferentiated cells occur, whereas MUC2 staining was in cells toward the centre and top of the crypts, where terminally differentiated cells are found (Fig. 4c). This suggests that ITLN1 and SPINK4 might mark an earlier stage of goblet cell differentiation than MUC2 in the intestine.

In the three patients with OSGs found in the 140 squamous endoscopic biopsies from 80 patients with BO, we observed that OSG cells consistently co-expressed ITLN1, and MUC2, but not SPINK4. This may be because SPINK4 positive cells are more 'naïve' in goblet cell differentiation and thus they are present lower in the duct or gland and were not captured within these biopsies (Fig. 4d). In these same three patients we found a squamous marker (KRT14, which pairs with KRT5 in p63 + cells), a columnar marker (KRT7) and a specialised goblet cells marker (MUC2) expressed in adjacent cells in the same OSG (Fig. 4e). This intestinal metaplasia in an OSG from a squamous oesophageal biopsy 20 mm proximal to the BO margin suggests the ability of OSGs to undergo intestinalisation and may be the source of BO islands[32]. In 30 BO endoscopic mucosal resection (EMR) specimens (from 16 patients) with intestinal metaplasia but no dysplasia present, we also consistently observed cells expressing ITLN1 or SPINK4 without MUC2 (Fig. 4f, also see Supplementary Table 2). Specifically, quantification of triple immunofluorescence staining of eight BO EMR specimens with intestinal metaplasia but no dysplasia taken from five patients showed 41% of MUC2 low cells expressed SPINK4 and/or ITLN1, whereas 28% of cells expressed MUC2 alone (Supplementary Table 3). These data suggest that OSGs and BO may contain early goblet cells, as seen in the colon, and that ITLN1 or SPINK4 might mark cells with some goblet cell characteristics that are not yet morphologically identifiable as goblet cells.

**OLFM4 marks a stem-like transcript profile in BO and OSG epithelium**. StemID is a published workflow designed to find cells with stem-like properties in single cell RNA-seq data by calculating a 'stem-ness' score based on the entropy of cell clusters and the number of links between clusters[33,34]. As a control we analysed duodenum cells from BO patients (A-D) and found the highest scoring cluster was enriched for *LGR5* expression, consistent with *LGR5* being a known marker of intestinal stem cells[35,36]. Applying StemID to the remaining individual tissues from the same patients did not identify any well-known stem cell markers (Supplementary Fig. 8a-b), even though a small number of LGR5 positive cells are present in all tissues sequenced (Supplementary Fig. 1). Since a recent study showed that BO contains pluripotent cells[37] and in view of the striking transcript profile overlap between OSG and BO cells, we therefore analysed all BO

and OSG cells using StemID (patients A–F). Interestingly, the highest scoring cluster was enriched for the stem-cell associated gene *OLFM4* (Fig. 5a, blue asterisk). BO cells from all four patients with BO (A–D) contributed to this cluster, and oesophageal cells from two patients with BO (A and B)

(Supplementary Fig. 8c). The second highest scoring cell cluster (Fig. 5a, red asterisk) was enriched for *LYZ*, a marker of Paneth cells, which are long-lived secretory cells found adjacent to the stem cell niche in the intestinal crypt base. *OLFM4* has been shown to associate with *LGR5* expression and marks stem cells in

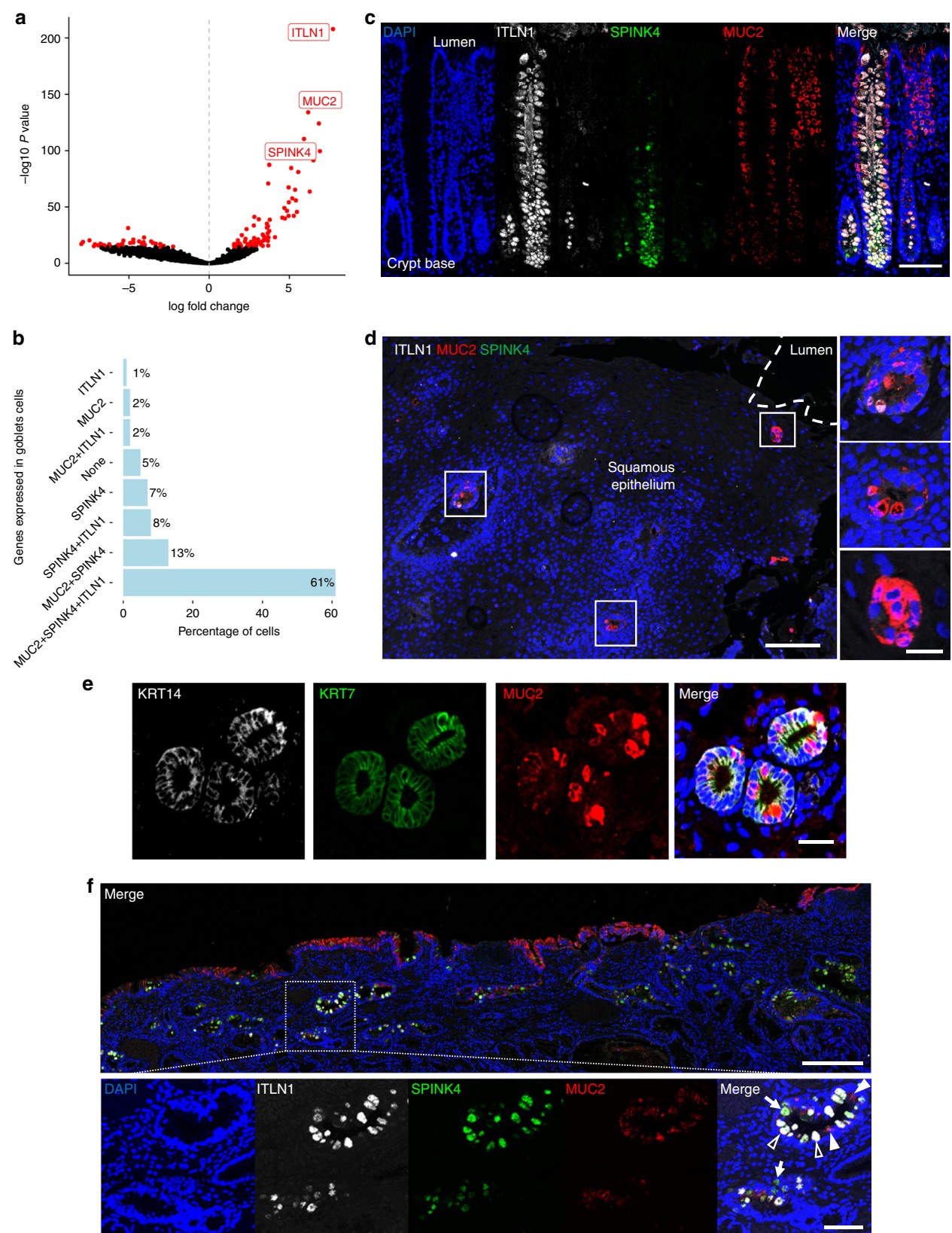

**Fig. 4** SPINK4 and ITLN1 mark early goblet cells. **a** Volcano plot showing fold change and *P* value of genes differentially expressed in the 'goblet-type' cell cluster as compared to all other cell clusters (see Fig. 3). Points coloured red indicate genes significant at 5% permutation test. Selected highly significant genes are labelled. **b** Bar chart showing the percentage of cells in the 'goblet-type' cell cluster (*n* = 98) expressing *MUC2*, *ITLN1* or *SPINK4* alone or in different combinations (thresholds set at the tenth centile to include 90% of cells in which at least one transcript was detected from each gene). **c** Triple immunofluorescence staining images of MUC2 (red), ITLN1 (white) and SPINK4 (green) in normal colon from a resection specimen (blue stain is DAPI). Scale bar, 100 μm. **d** Triple immunofluorescence staining images of MUC2 (red), ITLN1 (white) and SPINK4 (green) in normal oesophageal epithelium obtained by endoscopic biopsy (blue stain is DAPI). OSGs encroaching on the surface epithelium are shown in the enlarged images on the right. Scale bars are 200 μm and 50 μm in enlarged images. **e** Triple immunofluorescence staining images of KRT14 (white), KRT7 (green) and MUC2 (red) in an OSG beneath normal squamous epithelium from an endoscopic biopsy of normal squamous epithelium from a patient with BO biopsy (blue stain is DAPI). Scale bar 50 μm. **f** Representative immunofluorescence staining of Barrett's EMR specimen containing intestinal metaplasia but no dysplasia for MUC2 (red), ITLN1 (white) and SPINK4 (green); nuclei (DAPI) in blue. Scale bars are 400 μm and 100 μm in enlarged images

intestinal tissue in normal and metaplastic contexts[38,39]. Consistent with this, immunohistochemical staining detected OLFM4 expression in human colon crypt bases, where stem cells are known to be located (Fig. 5b). In 8 BO sections from 7 patients, we observed that OLFM4 protein expression was less restricted to the crypt base (Fig. 5c), similar to previous observations of LGR5 expression patterns in BO[12] and in contrast to the expression of OLFM4 in control tissues (Supplementary Fig. 8d). In OSGs beneath normal squamous epithelium, OLFM4 positive cells were seen within the gland structures (Fig. 5d). Interestingly, OLFM4 staining in OSGs from patients without BO was much more restricted than seen in OSGs taken from patients with BO (Fig. 5d, e), although the number of cases examined is limited.

Notably, *OLFM4* has a higher mean expression in the *LEFTY1*-positive clusters (B2/B3) compared to the clusters expressing known markers of the differentiated goblet (*MUC2*) and enteroendocrine (*CHGA*) lineages (Fig. 2a, B1 and B4, respectively). To examine co-expression of *OLFM4*, *LEFTY1*, *MUC2* and *CHGA* in individual cells, we applied a threshold at the tenth centile to include 90% of cells in which at least one transcript was detected from each gene of interest. Using this threshold, half of the BO cells express *LEFTY1* and *OLFM4*, alone or in combination (29% *OLFM4* and *LEFTY1*; 13% *OLFM4* only; 11% *LEFTY1* only). *LEFTY1* and *OLFM4* positive BO cells rarely co-expressed *MUC2* or *CHGA* (Supplementary Fig. 8e). Together, these data suggest that B2/B3 represent a cell population that harbours BO progenitor cells.

## Discussion

Our single cell RNA-seq data has resolved cell sub-populations in gastrointestinal epithelia and shown a profound similarity in the transcript profile between OSG cells and BO cells. This is supported by our observation that this sub-population of BO cells and OSGs express the stem cell-associated gene *OLFM4*, in line with the notion that these populations might contain similar progenitor cells. Glandular epithelial cells are replaced by squamous epithelium during development of the oesophagus and OSGs are functionally important structures formed from remaining glandular epithelium[40]. It is thus not surprising that the developmental gene *LEFTY1* is expressed in OSGs, and that as these structures expand during the development of BO, increased levels of LEFTY1 and OLFM4 are observed in these tissues. Notably, *LEFTY1* is regulated by TGF-β signalling and bone morphogenic proteins (BMPs)[41,42]. Since TGF-β is often perturbed in BO, and BMPs have been shown to play a major role in the development of a BO like phenotype, it will be interesting to explore these relationships further[43,44].

Additionally, our findings support a previously proposed hypothesis that BO may originate from OSGs. This model suggests that acid and bile reflux-induced damage to the oesophagus is 'repaired' by the expansion or selection of OSGs, which contain progenitors that may express OLFM4 and have alkaline

secretions, and are thus able to play a role in protecting the oesophagus from gastro-oesophageal reflux damage. Further consideration of the functional overlap of other secretory structures with BO and OSGs, such as salivary and mammary glands may help our understanding of an adaptive response to injury that drives metaplasia. Studies are also needed to experimentally demonstrate the potential of OSG cells, p63[+] or p63[−] OSGs in particular, to develop into BO cells and OAC.

Given that rodents lack OSGs, and the lack of an in vitro model of human oesophageal glands, analysis of human biopsies currently provides the most reliable approach to dissect the cell relationships of BO. Future improvements in single cell sequencing techniques may enable more systematic genetic confirmation of the cellular origin of BO through DNA analysis and also allow higher throughput, to reduce any potential selection bias inherent in the methodology we have used, especially with respect to gastric cells, which were likely to have been detrimentally affected by acid exposure. Also, it is important to note that our study cannot definitively identify the origins of OAC. Future studies are needed to address the relationship between BO and OAC on a cellular level, and how this relates to recent work suggesting that OAC is highly similar to a sub-set of gastric cancers[45].

Finally we showed that SPINK4 and ITLN1 seem to identify an earlier stage of intestinal metaplasia than marked by MUC2, given that they are expressed lower in intestinal crypts than MUC2 and can be seen without MUC2 in BO. Of clinical importance, our results suggest that intestinal goblet cell characteristics exist even in the absence of morphologically identifiable goblet cells, supporting the view that diagnosis of BO should not require the detection of goblet cells. Together, our findings help characterise BO in humans. In addition, this study demonstrates the power of single cell analysis of clinical samples to uncover biological relationships among cell types and cellular heterogeneity in healthy and diseased tissues.

## Methods

**Sampling.** Patients attending routine endoscopic surveillance of BO and patients with mild reflux symptoms undergoing gastroscopy for diagnostic purposes gave written informed consent and provided samples (patients A-F and I-Q, study authorised by South Central - Oxford C Research Ethics Committee: 09/H0606/5 + 5; patients G-H, study authorised by Yorkshire & The Humber - Sheffield Research Ethics Committee: 16/YH/0247). Patient numbers were chosen to provide suitable biological replicates, and cells sequenced to provide balanced sample sizes at sequencing input. Double bite quadratic 2 mm biopsies were obtained endoscopically using standard biopsy forceps (Radial Jaw 4 Standard Capacity, Boston Scientific, Natick, USA) from a central region of the BO segment avoiding the proximal BO margin as well as the oesophagogastric junction. Control samples were taken from the second part of the duodenum, the stomach 20 mm distal to the gastro-oesophageal junction and the normal oesophageal squamous epithelium at least 20 mm clear of the most proximal extent of BO. Each sample was fragmented and then pooled to ensure all sampling sites were represented in each investigative modality. Fragments pools were divided into three groups for histological verification, whole-tissue RNA-seq and single cell RNA-seq (Fig. 1a). Patients were selected based on their previously known pathological features (Supplementary Table 1 and Supplementary Fig. 9). Patients without BO described 0–2 reflux

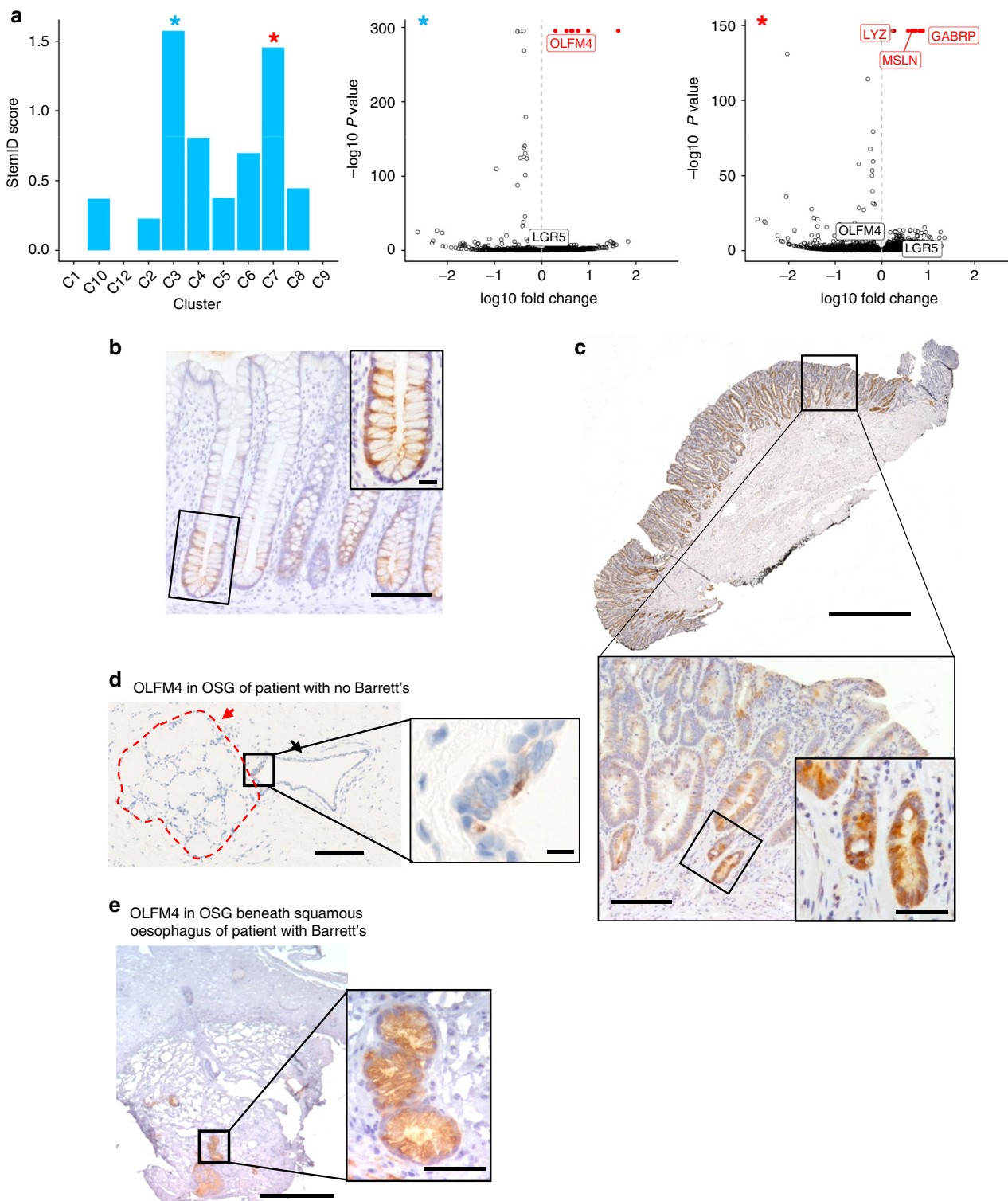

**d** OLFM4 in OSG of patient with no Barrett's

**e** OLFM4 in OSG beneath squamous oesophagus of patient with Barrett's

episodes per week with normal endoscopic appearances of the upper gastro-intestinal tract on endoscopic examination, and no histological evidence of oeso-phagitis in the processed samples.

**Cell isolation**. Sample fragments were placed directly into a digestion solution (made with 1x phosphate buffered solution (Gibco™), 2 mM EDTA, 100 U ml⁻¹ type I collagenase (Worthington Biochemical Company®), sodium phosphate (5.6 mM), monopotassium phosphate (8 mM), sodium chloride (96 mM), potassium chloride (1.6 mM), sucrose (44 mM), D-Sorbitol (55 mM), Dl-Dithiotreitol (0.5 mM)) and gently oscillated at 4 °C for 60 min. Samples were then further

fragmented with scissors and briefly manually triturated with a p1000 pipette. Fragments were allowed to settle and the cell-containing supernatant filtered (Sysmex Celltrics® 100 micron) into a 15 ml Falcon tube. This process was repeated 3 times and the product centrifuged at 300 g for 20 min at 4 °C to create a cell pellet which was resuspended in sorting buffer (1x phosphate buffered solution (Gibco™), 2 mM EDTA and 5% heat inactivated foetal bovine serum (Sigma-Aldrich®)). A small amount of each sample was pooled for labelling controls. Pre-conjugated CD45-FITC (1:10, mouse monoclonal, cat. 130-080-202, Miltenyi Biotec)[46] and EpCAM-PE (1:10, mouse monoclonal, cat. 130-110-999, Miltenyi Biotec)[47] antibodies were added to cell suspensions to help identify epithelial and immune cells, respectively, and they were incubated/washed according to manufacturer's advice.

**Fig. 5** *OLFM4* is upregulated in BO and OSG cells with stem-like transcript profiles. **a** Bar plot on left shows StemID scores across all RaceID2 clusters (see Methods) applied to all non-squamous oesophageal cells (BO and oesophageal cells with <5 KRT14 counts to exclude squamous cells, n = 533). Scores are calculated from multiplication of the entropy (spread from the cluster mean) and the number of cluster links arising from a given cluster. Differentially expressed genes in the highest scoring cluster (C3, blue asterisk) and second highest scoring cluster (C7, red asterisk) are shown in the volcano plots in the centre and right plots, respectively. Points coloured red indicate the most significant genes with a fold change >2. Selected highly significant genes are labelled. **b** Immunohistochemical staining of OLFM4 in human colon (close-up of base of crypt inset). Scale bars are 100 μm and 20 μm in inset. **c** Immunohistochemical staining of OLFM4 in BO mucosal resection containing intestinal metaplasia but no dysplasia, with enlarged image. Scale bars are 1000 μm, 200 μm in enlarged image and 50 μm in inset. **d** Immunohistochemical staining of OLFM4 in OSG under normal oesophagus taken from the proximal part of an oesophagectomy specimen resected for Siewert type III junctional tumour in a patient with no BO. Red dashed area and arrow indicates OSG, black arrow indicates OSG duct. Scale bars are 300 μm and 20 μm in enlarged image. **e** Immunohistochemistry in OSGs from endoscopic biopsy of normal squamous oesophagus in patients with BO. Scale bars are 300 μm and 50 μm in enlarged image

DAPI (1:2000, Sigma-Aldrich®) was added to cell suspensions immediately prior to sorting. FACS was carried out using a BD Biosciences FACS Aria IIIu platform with 70 μm nozzle in the case of the first four patients and the additional squamous samples, and a Sony SH800S Cell Sorter with 100 μm chip in the second batch of two patients including the pyloric samples. Cells were selected based on size and singlet gating to saturate cell output while minimising debris passed to subsequent gates. Size and singlet gating were then adjusted to capture of EpCAM+ cells, on the basis that these would represent a range of epithelial cells and minimise debris selection (Supplementary Fig. 10a). Resultant cells were sorted directly into 96 well plates (Life Technologies™ MicroAmp® Optical 96-well Reaction Plate) pre-prepared with 2 μl 0.2% Triton™ X-100 (Sigma-Aldrich®) and RNAse inhibitor (Takara Recombinant RNase Inhibitor) at 19:1 and then immediately frozen on dry ice. To confirm spectral accuracy, compensation bead controls and pooled cell suspensions were used for fluorescence-minus-one controls where possible. Each plate was re-permuted to avoid batch effects at the next stages of preparation, with no single plate containing cells from only a single patient or tissue type. Variable patterns of 6 blank wells were also prepared in each plate, 3 of which had a 10 pg of brain total RNA (Agilent Technologies) added as a positive control. A single 100 cell pool was also sorted in experiments involving pyloric cells (patients G-H) to provide a bulk control as whole tissue RNA-seq was not performed in these patients. To check for bias in cell selection, index sorting was carried out in most experiments to analyse expression of antibodies in relation to tissue type and subsequent data quality (Supplementary Fig. 10b-d). Using the input metrics available up to the point of sequencing, logistic regression was also undertaken to see if higher quality cell data could be predicted before sequencing. While the length of the experiment tended towards having an effect on data quality, recorded metrics at FACS could not accurately predict whether a cell would meet a read count threshold (Supplementary Fig. 10d).

**Single cell RNA-seq.** Transcriptome libraries were prepared using a Biomek FX liquid handling instrument (Beckman Coulter) with a custom adaptation of the published smart-seq2 method[48,49], with minor modifications, and Nextera XT (Illumina®) methodology with custom, unique index primers after tagmentation and ERCC spike-in at a dilution of 1:100,000. Libraries were sequenced using the Illumina® HiSeq 4000 platform, aiming for $3.5 \times 10^5$ reads per cell at 75 bp paired end.

**Bulk RNA-seq.** Tissue fragments were processed using the *mir*Vana™ miRNA Isolation Kit (ThermoFisher) according to manufacturer's guidance. Total RNA was enriched using ribodepletion (Ribo-Zero, Illumina®) prior to cDNA conversion. Second strand DNA synthesis incorporated dUTP. cDNA was end-repaired, A-tailed and adaptor-ligated. Samples then underwent uridine digestion. The prepared libraries were size-selected and multiplexed before 75 bp paired end sequencing using the Illumina® HiSeq 4000 platform.

**Data analysis.** All data were mapped using STAR[50] (release 2.5.2a) to the hg19 version of the human genome with transcriptome annotations from Gencode (release 25). Counts tables were made with HTSeq[51]. Cells were excluded that did not meet a threshold set to exclude all negative controls and outliers, and includes all remaining positive controls, see Supplementary Fig. 11a-c). For example, this was fewer than 25,119 fragments mapping to the transcriptome in the first experiment (patients A-D). No oesophageal cells from patient C passed this quality control threshold. To check biological relevance, counts from the most abundant cell population from a single patient and tissue were summed and correlated against bulk RNA-seq expression (Supplementary Fig. 11d). Counts were trimmed mean of M-values (TMM)-normalised and fragments per kilobase million (FPKM) values were calculated. Genes with less than 4 FPKM in at least 3 cells were filtered out. After re-normalisation, expression values were converted to transcripts per kilobase million (TPM). A further gene filtering step was included to remove highly expressed genes with low variability across all samples (cells in the top decile for mean expression and below the fifth centile for coefficient of variation). SC3[52] was

used to provide cell cluster information. Cluster robustness to experimental technical variation was tested using BEARscc[28] which models technical noise from ERCC spike-in measurements. Cluster number, *k*, was chosen manually using the distribution of cluster-wise mean silhouette widths across clusters in all 250 simulated technical replicates for each cluster number *k* (2–8 for individual tissue and 1–15 for all tissues). Where box plots are used, the lower and upper hinges correspond to the first and third quartiles (the 25th and 75th percentiles), the whiskers extend from the hinge to the largest or smallest values at most 1.5x inter-quartile range from the hinge. Data beyond the whiskers are outliers and are plotted individually. t-SNE data were generated using the Barnes-Hut implementation of t-SNE[53] in R. Differential expression analysis was carried out between cell groups using edgeR[54] from normalised counts according to the package manual. *P* values used were determined by permutation test at 5% (250–1000 permutations) to allow for multiple comparisons or, in cases of unbalanced sample numbers, converted to false discovery rates (FDR) by the Benjamini-Hochberg procedure. Pathway analysis was performed using goseq[55] to identify over or under represented ontological terms. Identification of stem-like cells was performed using RaceID2 and StemID, please see https://github.com/dgrun/StemID for more details[33,34]. Further results from this analysis showing differentially expressed genes in high stem-scoring clusters are available in Supplementary Data 4. Where gene expression is described in binary terms, the threshold was set to include or exclude 90% of cells with the highest expression of a given gene, to allow for biological noise.

**Immunohistochemistry and immunofluorescence on human tissue.** Oesophageal samples from oesophagectomy specimens (5 patients) containing normal mucosa and gland structures and endoscopic mucosal resection specimens (30 patients) with Barrett's oesophagus were obtained from the Oxford Radcliffe and Translational Gastroenterology Unit biobanks. Sections were de-waxed, rehydrated, and incubated with 3% hydrogen peroxide in methanol to block endogenous peroxidase activity (10 min, room temperature). Antigen retrieval was carried out using 10 mM sodium citrate, pH6 at 100 °C for 10 min. Sections were then blocked with normal goat serum (at room temperature) and incubated overnight at 4 °C with a primary antibody against anti-KRT14 (IHC, 1:1000, rabbit polyclonal, cat. PRB-155P, BioLegend), anti-TFF3 (IHC, 1:1000, mouse monoclonal, cat. WH0007033M1, Sigma-Aldrich®)[56], anti-MUC2 (IHC, 1:300, rabbit polyclonal, cat. SC-15334, Santa Cruz Biotechnology)[57], anti-CHGA (IHC, 1:500, rabbit polyclonal, cat. ab15160, Abcam)[58], anti-KRT7 (IHC, 1:4000, rabbit monoclonal, cat. ab181598, Abcam)[59], anti-LEFTY1 (IHC, 1:1000, D7E3G rabbit polyclonal, cat. 12647, Cell Signalling), anti-OLFM4 (IHC, 1:200, D1E4M rabbit monoclonal, cat. 14369, Cell Signalling Technology®), anti-ITLN1 (IHC/IF, 1:500, sheep polyclonal, cat. AF4254, R&D systems)[60], anti-MUC2 (IF, 1:300, mouse monoclonal, cat. ab11197, Abcam)[61] or anti-SPINK4 (IF, 1:500, rabbit polyclonal, cat. HPA007286, Sigma-Aldrich®)[62]. For immunohistochemical staining, samples were then treated with biotinylated secondary antibody (Vector Labs; 1:250) for 40 min at room temperature. The staining reaction was worked up using the Vector Elite ABC kit and counterstained with haematoxylin. Samples were examined by a pathologist using a histology microscope. For immunofluorescent staining, expression was detected using Alexa Fluor (1:250, Molecular Probes) for one hour. DAPI (1:2000, Sigma-Aldrich®) was used to stain nucleic acids. Samples were observed using a confocal microscope system (LSM 710; Carl Zeiss). The limited amount of material obtained from patients precluded the use of each described staining technique on every sample collected.

## Data availability

Single cell and bulk RNA-seq counts data and the cell cluster assignments for each analysis are supplied in the Supplementary Data Files 5–7. Raw data are available in the European Genome-phenome Archive, following the necessary consents to protect donor anonymity (accession # EGAS00001003144). All other data available upon request.

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

## Acknowledgements

The work is mainly funded by the Ludwig Institute for Cancer Research Ltd. with additional support from a CRUK Accelerator Award (C328/A21998) and the NIHR Biomedical Research Centre. The views expressed are those of the authors and not necessarily those of the NHS, the NIHR or the Department of Health. R.P.O. received funding from the Oxford Health Services Research Committee and Oxford University Clinical Academic Graduate School. M.J.W. was supported by Cancer Research UK (C5255/A19498, through an Oxford Cancer Research Centre Clinical Research Training Fellowship). D.T.S. was supported by the Nuffield Department of Clinical Medicine and the Clarendon Fund. C.P.P. was funded by an MRC programme grant (MC_UU_00007/15). We thank Mary Muers and Françoise Howe for helping with critical reading of the manuscript, Andrew Roth for comments on statistical analysis of the data, Sally-Ann Clark and Paul Sopp for providing FACS expertise, John Findlay for help with ethical approval, and Rory Bowden, Amy Trebes and the High-Throughput Genomics team at the Wellcome Trust Centre for Human Genetics, Oxford for assistance with sequencing.

## Author contributions

R.P.O. and M.J.W. collected biopsy samples and prepared them for sequencing. M.J.W. carried out the immunoreactive staining and imaging. M.J.W. and C.R.-P. processed the FFPE samples. R.P.O. and D.T.S. carried out RNA-seq mapping and data analysis. B.B., A.B., M.R.M. and N.D.M. helped to design and curate the clinical data and sample collection. R.G. and L.M.W. provided pathological interpretation of all samples used. A.G., P.P. and D.B. generated all sequencing data used. C.P.P. provided computational oversight of the data analysis. B.S.-B. provided overall supervision of the computational analysis of the data and X.L. provided overall supervision of the project. The manuscript was written by R.P.O., M.J.W. and X.L., with assistance from B.S.-B. and D.T.S. Figures were prepared by R.P.O., M.J.W. and D.T.S.

## Additional information

**Competing interests:** The authors declare no competing interests.

