## [Peer Review File · Nature Communications]

Reviewers' Comments:

Reviewer #1:

Remarks to the Author:

The paper entitled, 'Single cell RNA-seq reveals profound transcriptional similarity between Barrett's esophagus and esophageal glands', by Owen et al., gives an overview of the transcriptome of epithelial cells from the upper Gastro-intestinal tract including Barrett's esophagus. The authors performed single cell RNA sequencing using biopsies from patients. The authors report as their most interesting finding the fact the LEFTY1 and OLFM4 are found to be expressed in a subset of the single Barrett cells. Also, they found a similarity between cells that they refer to as ESGC (I presume that these are commonly named: esophageal submucosal glands) and the Barrett cells.

The article gives an overall descriptive overview of epithelial transcriptomes, more clarity is required regarding the methodology and interpretation of the data.

Major comments

1. The analysis is based on material of 4 Barrett patients and the number of cells that each patient contributed is not clear. In the abstract the authors report to have analysed 2895 cells while in the article the number of analysable cells was only 1778. The number of cells in the abstract need to be adjusted (or left out), and the number of Barrett cells that were finally analysed from each patient need to be more clearly mentioned in the text.

2. The authors mention several times that they used 'uncharacterised Barrett cell populations', However, in the method section it seems that cell have been isolated based on Epcam positivity and in their methodology for single cell RNA sequencing, only viable cells (DAPI negative cells) could be used for the RNA sequencing analysis. This introduces several biases in the selection of the cell populations, which is of importance for the interpretation of the data. Although, many Barrett cells are Epcam positive, there is a subpopulation that do not carry this marker. How was this corrected? Also, it is to be expected that differentiated cells in Barrett villi may be less vital compared to proliferating cells in the glands, by selecting the more vital subpopulation, certain cell types can be easily missed. There is no mention about the possible biases and no correction for biases that the methodology may have introduced.

3. Which markers were used to select the squamous epithelial cells? Is Epcam a representative marker for these cells?

4. The authors classified a group of cells in the esophageal specimens that they found to be 'esophageal gland complexes (ESGC)', I presume also known as esophageal submucosal glands. Based on the expression pattern and validation of several markers (including TFF3) the authors found an overlap between these ESGC cells and Barrett cells. In figure 4 the authors point to similar glands in the Barrett biopsies. It is not clear from this figure whether these are sub-mucosal glands or the glandular region of a Barrett crypt. At least an H&E of a consecutive tissue slides needs to be joined to this panel. Also these esophageal glands are not as "rare" as mentioned by the authors. (The submucosal glands are as expected in the submucosa and therefore are rarely are seen in endoscopic esophageal biopsies). Nevertheless, the authors point to a rather large population of ESGC cells in the squamous samples. Given the possible bias introduced by the methodology, which at the end resulted in a rather small number of cells per biopsy (100-200 cells??), it is possible that in both the squamous and the Barrett sample were enriched for cells from the submucosal glands. This is in line with the fact that LEFTY1 and OLFM4 were found to be highly expressed in a relative high number of Barrett cells. Both markers are clearly also expressed in the ESGC. This to me is a major matter of concern and needs to be clarified (see 5). One question is if the ESGC were also seen in the corresponding slides of the biopsies used for the single cell analysis?

5. Another important point is that it has been noted that the ESGC comprise of different cell lineages, including cells that carry markers for squamous lineages. These squamous cells were even found to possibly repopulate and give rise to neo-squamous epithelium after ablation and endoscopic mucosal resections. The authors do not point to at all to these different lineages nor to these papers. By IHC the expression of the squamous markers should be clearly noticeable in these glands. The authors do not point to such expression for instance in figure 1E. To my opinion

the only way to define the subpopulation of cells in the ESGC or submucosal glands is by dissection these glands from oesophageal specimens and subjecting these for single cell analysis, without preselecting cells for Epcam. This will indicate the different cell lineages and from which lineage the Barrett cells may arise.

6. The authors found LEFTY1 to be highly expressed in a large proportion of Barrett cells. Given the selection method of the cells this finding needs to be interpreted with care.

7. The authors only briefly mention the role of LEFTY1 and its association as a TGF-beta family member. Although no extensive data has been published on LEFTY1 and Barrett's, there are several papers that indicate the role of several TGF-beta family members, for instance in the pathogenesis of Barrett's. Also, there are several papers mentioning the regulation of LEFTY1 by other TGF-beta family members. It would be of high interest to perform pathway analysis in order to better understand and demonstrate the functional role of LEFTY1 in signalling. The RNA seq data should easily allow such analysis.

8. Minor points:

Abstract

-The authors mention 'acid' as the sole component in the refluxate of Barrett patients. This is incomplete since both bile and acids can be found in the refluxates of Barrett patients. The role of bile in the pathogenesis of Barrett's has been described in several papers. This needs to be corrected.

-correct the number of cells or do not mention a number that is overstating.

-remove the word uncharacterized, since Epcam was used to sort the cells.

Introduction

Correct the statement 'highly fatal'. Give proper numbers for survival and a reference. Do not refer to review articles but to original papers (e.g., ref 9)

Result section

- It is mentioned that the normal patients had esophagitis. Was this also the case while taking biopsies?? If so this could have influenced the expression profiles of these cells, which needs to be addressed.

- Include how many cells per biopsy/patients were finally included in the analysis.

- On page 7, ESGC are mentioned as rare cells. These glands are present in the proximal and distal esophagus and are functional (not 'remnants') in secreting mucins to lubricate and protect the esophageal mucosa. This needs to be corrected in the statements the authors made in the discussion. The glands are only rare in biopsies, because of their submucosal location. This needs to be corrected, also in the discussion.

- Figure 2: the LEFTY1 staining is not convincing. Is the expression cytoplasmic or nuclear? Better stainings are required and if still not clear I would encourage RNA-ISH to better identify the expression and include overview of the stainings.

- Page 8. The authors state that 19% were (MUC2 positive) goblet cells. Did all patients had a number of goblets cells? Since in the material section it is stated that only patients with intestinal type of Barrett metaplasia were included, one would expect that all patients had a fraction of goblet cells.

- Figure 1 represents an oesophagus with an adenocarcinoma in which submucosal glands are presumed to be positive for TFF3. However, in this case there is also an adenocarcinoma expressing TFF3. To me the structures pointed out as ESGC could as well be adenocarcinoma cell clusters. Specimens that better represent the ESGC should be included.

Reviewer #2:

Remarks to the Author:

The main premise of this article is identification and characterisation of different cell types within Barrett's oesophagus (BE) and comparison of this tissue with other samples taken from GI tract. In order to study the samples, the authors used single cell RNA-seq, specifically the smart-seq2 methodology. Owen et. al. sequenced 2895 cells from 7 patients (5 patient with Barrett's oesophagus (BE) and 2 control patients) and identified a cell population within BE marked by

LEFTY1 and OLFM4, which overlaps a subset of oesophageal cells. A hypothesis proposed by the authors is that this novel population of cells originates from the submucosal glands of normal oesophagus, and the authors postulate that the submucosal glands are the site of origin for BE.

This is an interesting hypothesis, investigated with a substantial sequencing effort.

There are a number of major comments that need to be addressed in a revised version of this manuscript:

1) The concept that Barrett's originates from submucosal gland cells is not new and although the single cell RNA seq data is elegant, no definitive evidence is provided to confirm or refute this hypothesis. A direct comparison with the single cell gene expression profile obtained from a squamous submucosal esophageal glands is essential to test this hypothesis. Specific issues are:

- The authors state that expression of TFF2 and TFF3 in samples from squamous esophagus are consistent with these cells being from esophageal gland complexes - and this is reasonable. However, these trefoil factors are widely expressed in other parts of the GI tract, and therefore this information is not on its own sufficient to determine whether or not Barrett's arises from submucosal gland ducts.

- The authors use a method of elimination to deduce that a subset of BE epithelial cells have a gene expression profile consistent with their being ESGCs (p9, Fig3c and suppl Table 2). The comparison between BE cells with an ESGC profile and the transcriptional relationship of cells from other tissues (p9, last para) is helpful but not definitive.

This needs to be clarified in the presentation of the manuscript.

2) It is unclear how many cells were included for each analysis. Furthermore, it is not entirely clear how cells from each patient contributed to the results. Please make this information transparent, since it is important for determining whether interpatient batch effects are contributing to the clustering. Based on our analysis of the supplementary data used for clustering in the Figure 1, it seems as though the conclusions reached depend on a small number of cells arising almost entirely from a single patient.

Based on our understanding of the clusters, we conclude that:

- The TFF2/TFF3 high cells derive almost entirely from Patient A.
- For the esophageal data patients B and D have almost no cells in cluster 1. Patient C had no esophageal data. One would expect them all to have expression confirming the esophageal origin.
- Data for patient A indicate that around 50% of the esophageal samples (either within a single biopsy or 50% of individual biopsies) fall into the TFF2/TFF3-high group. Presence of such a large population of cells should be observed within the bulk RNA-seq data. This is not observed in figure 2c
- The gastric profile varies enormously between patients. The duodenal samples have a much better representation across the clusters suggesting that the data are comparable.
- As a quality check, it might be beneficial to compare the bulk RNA-seq from patient A with the combined scRNA-seq data for that patient, in order to confirm that the average scRNA-seq signal is similar to bulk RNA-seq

3) The markers of "early goblet cells" are correlative and not conclusive. Functional studies would be required to take these cell lineages and show that they can differentiate into goblet cells.

4) The authors sought a stem-like population in the BE and ESGC cells by applying an algorithm to the single cell RNA profile. The highest scoring cluster (C3 in Figure 5b) was predicted as a stem-like population, however it is not shown whether this stem-like population is found in both BE and

ESGC cells, or in all patients. It raises the same issue as the aforementioned ESGC cells, and appears to be another n=1 case.

5) The clinical implications of the findings are over-stated. Barrett's is a straightforward diagnosis made from endoscopy and histopathology, and molecular analysis is not required, and indeed would add considerable financial and logistical burden. The value of molecular analyses maybe to help differentiate those patients at high risk for cancer progression, but this is not the topic of this manuscript.

6) The authors state that this study does not provide data relevant to cancer progression in Barrett' and that further studies will be required to inform on this aspect. However, 8/19 samples were taken from BE with dysplastic foci (Extended Table 2). Therefore, it is not clear whether some of the transcriptional changes do in fact reflect the progress towards cancer.

7) The introduction states that the current most widely held view is that Barrett's originates from the stomach. This has been a controversial topic undergoing research for over 20 years with a number of competing theories, and it is not clear from the publications that there is a current consensus. The theory that Barrett's originates from submucosal glands of the squamous esophagus is not new; however the authors have neglected to cite the previous research about the submucosal gland duct as a possible source of origin for Barrett's. The previous data should be mentioned (see recent review on Origin of Barrett's epithelium in Cellular and Molecular Gastroenterology (Ref 1 below) and Hepatology (Ref 2 below) published which summarises much of the published evidence).

8) As noted in the introduction of the manuscript, a number of gastric sites have features recapitulated in Barrett's epithelium, and indeed the title of ref 11 is "gene expression profile of Barrett's epithelium replicates pyloric-type gastric glands". Therefore, the authors need to justify why they only sampled the gastric cardia region for comparison. In line with the hypothesis to "clarify the relationships between cells in normal tissues and BE" a wider upper GI sampling protocol would have allowed more robust conclusions to be drawn. This needs to be clarified in the presentation of the manuscript.

9) Please can the author clarify from where normal squamous oesophageal biopsies were taken from? This is important, as the density of submucosal glands varies according to the proximal-distal extent of the oesophagus. It appears that squamous samples showing a submucosal gland profile were taken from above the Barrett's- depending on the level in that patients the density of submucosal glands might be expected to be very low. This would also beg the question as to why the Barrett's did not extend higher if a submucosal gland was present at this location.

10) As above, clarification is needed to understand which samples were used for which experiment. When comparisons were made between transcriptional profiles of Barrett's and squamous samples was this between squamous epithelium of control patients only or Barrett's and controls? For Barrett's patients the normal squamous was taken from the proximal oesophagus above the Barrett's, and presumably in controls the squamous sampling was performed closer to the gastro-esophageal esophageal junction.

11) Hierarchical clustering showed that Barrett's samples were more closely related to gastric tissues. Was any attempt made to quantify the amount of IM present, since focal versus diffuse intestinalisation will likely affect this result?

12) Which other developmental genes were expressed in Barrett's versus other tissues? For example, HOX gene profiles have been interrogated in Barrett's previously with some evidence that the profile might most closely resemble HOX expression patterns observed in colonic tissues. In the Supplementary Table 2 it seems that HOXB6 is detected - is this correct? An explanation would be helpful.

13) It is not clear to me why the single cell RNAseq data on p7. show that “cell subpopulations, including rare subpopulations from ESGCs that cannot be identified by conventional RNA-seq”. We have known for years from H&E analysis of paraffin sections about submucosal glands in the human distal esophagus. The acute phase response genes are interesting but the significance of these findings is not clear.

14) If MUC2 marks goblet cells why were these present in the gastric samples (p10, Fig 4)? Whether ITLN1 and SPINK4 mark early goblet cells is an interesting idea, but it is not proven by this correlative analysis.

15) The volcano plot in Figure 5c (below) needs clarification. OLFM4 is selected as the stem-like cell marker because of its high expression in the highest-scored cluster C3 and its association with classic stem cell marker LGR5. However, the plot shows 5 red dots, does it mean there are another 4 genes similarly expressed as OLFM4?

16) The faint staining of OLFM4 in ESGC looks non-specific (see also minor comment below). Because this study does not provide functional or clonal assays, which are normally required for stem cell science, it is important to show clear expression patterns of the markers in tissue, as well as co-expression of classic stem cell markers/associated genes.

17) The study then focused on the co-expression pattern of OLFM4 and LEFTY1 in two subsets of BE cells i.e. B2 and B3 in Figure 2a. The representation of the stem-like population in ESGC cells from normal esophagus is not mentioned.

18) The subtitle: “OLFM4 marks stem-like transcriptional behaviour in columnar esophageal epithelium” (Line 221) is over-stated as the evidence mainly supports OLFM4 expressing stem-like subset residing in BE cells but not ESGC cells.

19) Given that another widely held theory is that BE originates from gastric stem cells, it is a little odd that this study did not also use the StemID algorithm to analyse gastric cells.

20) Extended data 8a: The H&E images are poor quality and show the surface of the epithelium and need higher power.

Minor comments:

- Abstract: Barrett's occurs in response to efflux of acid and bile, not just acid
- Results: The % of BE cells expressing LEFTY1 is given but no % for other genes listed. This should be added to enable comparisons to be made.
- KRT7 expression is noted as being “consistent with its clinical utility in BE diagnosis”. This is incorrect - KRT7 expression is not used clinically for Barrett's diagnosis.
- The origin of colonic samples should be stated in methods and ethical approvals.
- The ESGCs could be viewed as a developmental remnant, however they do have a physiological role in mucous production to protect the squamous epithelium.
- Fig. 4e: the BE glands look like epithelial glands rather than submucosal esophageal glands that can also sometimes be seen beneath Barrett's epithelium.
- Fig. 5: Stem-lie should read stem-like. The OLFM4 staining in ESGD looks like nonspecific background staining (image 5f).
- Extended data Table 2. The F:M ratio is surprising for this disease
- Methods. How did you define “mild reflux” for your control patients?
- Ref 2 is incomplete and the link or detailed source should be given
- Supplementary tables and material require legends and detailed description of content
- The ordering of clusters in the supplementary material is different to the main figures
- High quality images are require for figures 1e, 2b, 4c-e, 5e,f, S8a
- Methods section refers to figure S6, we think it should actually be S8 (lines 525, 544, 571)

- Line 297 – most of the article is written in American English, however 'colours' uses British spelling
- There is not information provided on whether raw data will become available
- Methods: Please include the concentration of ERCCs added to the single cell RNAseq samples.

References

1. Garman, K. S. Origin of Barrett's Epithelium: Esophageal Submucosal Glands. *Cell. Mol. Gastroenterol. Hepatol.* 153–156 (2017). doi:10.1016/j.jcmgh.2017.01.016
2. van Nieuwenhove, Y., Destordeur, H. & Willems, G. Spatial distribution and cell kinetics of the glands in the human esophageal mucosa. *Eur. J. Morphol.* 39, 163–8 (2001).

Reviewer #3:

Remarks to the Author:

Barrett's oesophagus is both an important clinical problem and a fascinating phenomenon. Although the nature of the epithelial changes has been hotly debated, no clear concept has emerged. Against this background Owen and co-workers present an scRNAseq analysis that provides compelling evidence to suggest that Barrett's oesophagus may arise from the cells of the oesophageal gland.

I am not an expert in scRNAseq analysis, but I believe the work has been carried out to a high standard. The validation experiments, involving mapping expression of LEFTY and other hits, are very strong. The paper is very clearly written.

For me, this work represents a beautiful example of the power of scRNAseq to overturn long-held opinions about tissue function and its link to disease. The findings may ruffle a few feathers but I think the paper will attract widespread interest.

Reviewers' comments:

Reviewer #1 (Remarks to the Author):

NB. Figures commented in this reply are attached at the bottom of the document in the order they are referred to in this reply (though not duplicated). All figures have been stylistically updated since the last submission. Additional figures to address specific reviewer points are inserted in-line in the response.

The paper entitled, 'Single cell RNA-seq reveals profound transcriptional similarity between Barrett's esophagus and esophageal glands', by Owen et al., gives an overview of the transcriptome of epithelial cells from the upper Gastro-intestinal tract including Barrett's esophagus. The authors performed single cell RNA sequencing using biopsies from patients. The authors report as their most interesting finding the fact the LEFTY1 and OLFM4 are found to be expressed in a subset of the single Barrett cells. Also, they found a similarity between cells that they refer to as ESGC (I presume that these are commonly named: esophageal submucosal glands) and the Barrett cells.

Thanks for the comments. We have changed our nomenclature to 'oesophageal submucosal glands' or 'OSGs' throughout the manuscript and this document.

The article gives an overall descriptive overview of epithelial transcriptomes, more clarity is required regarding the methodology and interpretation of the data.

Major comments

1. The analysis is based on material of 4 Barrett patients and the number of cells that each patient contributed is not clear. In the abstract the authors report to have analysed 2895 cells while in the article the number of analysable cells was only 1778. The number of cells in the abstract need to be adjusted (or left out), and the number of Barrett cells that were finally analysed from each patient need to be more clearly mentioned in the text.

To simplify the message we have now omitted the number of cells sequenced in the abstract. During the revision we have added data from additional patients and have re-analysed the data with a defined criteria and have reached the same conclusion as that reported previously. All these are now clearly stated in the text and fully summarised in new **Extended Data Figure 11a**.

The text now states "A total of 4237 cells were sequenced from 8 patients (Extended Data Table 1) in three batches. Due to known issues with batch effects in single cell experiments²⁵, analysis of cells from each batch has been kept separate where feasible and cells were permuted across plates and pooled prior to sequencing (see Methods). The first batch yielded 1040 cells (207 duodenum, 227 gastric, 371 BO and 235 oesophagus) suitable for analysis from four patients (A-D) with BO and intestinal metaplasia. A total of 214, 35, 66 and 56 BO cells were analysed from each BO patient, respectively. The second batch yielded 648 oesophagus cells suitable for analysis from two patients (E-F) with symptoms of gastro-oesophageal reflux but no identifiable oesophageal pathology. Finally, the third batch of cells yielded 194 cells (29 pylorus, 109 gastric, 32 BO and 24 oesophagus) suitable for analysis from two patients (G-H) with BO and intestinal metaplasia. Overall, a mean of 1.2×10^5

reads were mapped per cell and a median of 3978 genes were detected per cell (with at least one read per gene).”

2. The authors mention several times that they used ‘uncharacterised Barrett cell populations’, However, in the method section it seems that cell have been isolated based on Epcam positivity and in their methodology for single cell RNA sequencing, only viable cells (DAPI negative cells) could be used for the RNA sequencing analysis. This introduces several biases in the selection of the cell populations, which is of importance for the interpretation of the data. Although, many Barrett cells are Epcam positive, there is a subpopulation that do not carry this marker. How was this corrected?

Also, it is to be expected that differentiated cells in Barrett villi may be less vital compared to proliferating cells in the glands, by selecting the more vital subpopulation, certain cell types can be easily missed. There is no mention about the possible biases and no correction for biases that the methodology may have introduced.

We agree with the reviewer that the FACS sorted cells used in this study are biased towards cells that able to resist the harsh treatment of disaggregation and sorting and remain viable. To our knowledge, all single cell sequencing methods are biased towards viable cells when using fresh primary tissues. Therefore the term “uncharacterised Barrett cell populations” needs further clarification. We have amended the text to acknowledge this source of bias. However it is important to stress that all tissues used in this study were treated under the same conditions. Therefore they are subjected to the same bias thus the observed transcriptomic variations among different cell populations represent true differences. We have now also clarified the use of EpCAM (epithelial cell adhesion molecule). EpCAM expression has typically associated with cell proliferation and inversely associated with differentiation (Münz et al. The carcinoma-associated antigen EpCAM upregulates c-myc and induces cell proliferation. *Oncogene*, 2004; Barker et al. Identification of stem cells in small intestine and colon by marker gene Lgr5. *Nature*, 2007). In normal tissues, EpCAM is mainly expressed at basolateral cell membrane of simple epithelial cells. It is also found in pseudo stratified and transitional epithelial cells but not in differentiated squamous stratified epithelial cells. Consistent with this, EpCAM expression was not detected in oesophagus biopsies which only contained normal stratified squamous epithelial cells (top left image in **Extended Data Figure 10a** as an example) but it was detected in oesophageal submucosal glands or oesophageal submucosal gland duct cells. Moreover, its expression was also detected in the simple epithelial cells located in gastric, duodenum and BO biopsies (see new **Extended Data Figure 10a-b**). Interestingly the fluorescence expression level of EpCAM detected in BO epithelial cells during cell sorting is higher than that detected in epithelial cells present in all three adjacent normal controls. Since EpCAM expression is known to associate with epithelial cell proliferation rate, this indicates that cells in BO have higher proliferation rate than normal control epithelial cells (see new **Extended Data Figure 10c**, (Martowicz et al. EpCAM overexpression prolongs proliferative capacity of primary human breast epithelial cells and supports hyperplastic growth. *Mol Cancer*, 2013).

In this study, all cells obtained from every tissue biopsy were stained with EpCAM and CD45 antibodies. Cell sorting was based on size, doublet, CD45 and DAPI gating. Both EpCAM positive and EpCAM negative cells were sequenced. EpCAM was used to help adjust the cell sort by informing our size gating parameters to encompass most epithelial cell sizes and exclude debris. Due to the nature

of the tissues and the way the biopsies were taken, the majority of the cells from BO, gastric and duodenum were EpCAM positive epithelial cells. Nonetheless, cells were not collected using EpCAM as a selection marker, instead EpCAM was used to help guide cell size selection in forward and side scatter and to confirm their epithelial identity. As can be seen in extended data Figure 10b-c, the EpCAM expression profile is broad in each tissue, except normal oesophagus which is EpCAM negative in most cases, consistent with the immunohistochemical staining described above (**Extended Data Figure 10a**). We also used logistic regression to analyse the index sorting metrics we obtained for each sequenced cell to see if any of these metrics impacted on whether a cell was suitable for analysis (based on sequenced controls). **Extended Data Figure 10d** shows that the length of the sort and EPCAM expression had the most significant impact on sequencing success, but this was not very significant. The materials and methods section has been revised to reflect this additional analysis and clarify our cell selection.

3. Which markers were used to select the squamous epithelial cells? Is Epcam a representative marker for these cells?

We apologise for the confusion. We did not use any marker to select squamous epithelial cells. The cell sorting criteria were the same for all biopsies derived from all sites (see above). The text has been updated and **Extended Data Figure 10** (above) provides further details.

4. The authors classified a group of cells in the esophageal specimens that they found to be ‘esophageal gland complexes (ESGC)’, I presume also known as esophageal submucosal glands. Based on the expression pattern and validation of several markers (including TFF3) the authors found an overlap between these ESGC cells and Barrett cells the authors point to similar glands in the Barrett biopsies. It is not clear from this figure whether these are sub-mucosal glands or the glandular region of a Barrett crypt. At least an H&E of a consecutive tissue slides needs to be joined to this panel. Also these esophageal glands are not as “rare” as mentioned by the authors. (The submucosal glands are as expected in the submucosa and therefore are rarely are seen in endoscopic esophageal biopsies).

We have amended our nomenclature from Esophageal gland complexes (ESGC) to ‘oesophageal submucosal glands’ (OSG). We have replaced the image of an OSG under Barrett’s with another, clearer staining of KRT14+, KRT7+, MUC2 positive cells present in OSGs under proximal squamous epithelium found in 3 out of 140 biopsies from 80 patients with Barrett’s oesophagus (new **Figure 4e**).

With regard to the ‘rarity’ of OSGs, the reviewer is absolutely correct in that these are not rare in the human oesophagus, however they are rare in endoscopically obtained biopsies. To demonstrate this, we have examined a further 140 endoscopic biopsy samples taken from adjacent normal oesophagus (20mm proximal to BO) from 80 patients. (**Supplementary Table 2**) After sectioning each entire sample, we found that oesophageal submucosal glands were only detected in three different patients, two of which provided tissue sections of sufficient quality for use. Since OSGs are uncommon features in endoscopic squamous biopsies, we also examined OSGs in five oesophagostomy specimens obtained from patients with Siewert type III tumours below their

gastro-oesophageal junction, with no histologically detectable Barrett's oesophagus.

Nevertheless, the authors point to a rather large population of ESGC cells in the squamous samples. Given the possible bias introduced by the methodology, which at the end resulted in a rather small number of cells per biopsy (100-200 cells??), it is possible that in both the squamous and the Barrett sample were enriched for cells from the submucosal glands. This is in line with the fact that LEFTY1 and OLFM4 were found to be highly expressed in a relative high number of Barrett cells. Both markers are clearly also expressed in the ESGC. This to me is a major matter of concern and needs to be clarified (see 5). One question is if the ESGC were also seen in the corresponding slides of the biopsies used for the single cell analysis?

This reviewer's concern about the potential bias of the methodology used in this study has been addressed above (see reply to question 2). We would like to stress that all cells used in the single cell sequencing study were not selected based on EpCAM expression and cells from all tissues were treated under the same experimental conditions. Therefore the methods used in this study were not specifically set to enrich submucosal gland cells from squamous and Barrett samples. Additionally, our ability to detect differentiated cells such as enteroendocrine cells, and KRT7 expressing Barrett's cells suggests that submucosal glands were not exclusively sampled. OSG cells seen in the new data are beneath the squamous epithelium.

However like all single cell sequencing methods, it is biased towards viable cells. Therefore we cannot rule out the possibility that submucosal gland cells are more resistant to the experimental treatments required for single cell sequencing than other cell types. In fact this agrees with the notion that stem cells or progenitor cells are often more resistant to stress induced cell death than differentiated cells (Hoffman et al. Human embryonic stem cell stability. Stem Cell Rev, 2005). Thus the unintentional bias introduced by the single cell sequencing method may enable us to enrich cells with stem cell and progenitor property. Consistent with this, our study of LEFTY1 and OLFM4 expression in the OSGs of patients with and without Barrett's oesophagus shows that LEFTY1 and OLFM4 expression is sparse in the OSGs of patients without Barrett's. However, from 140 endoscopic biopsies of squamous oesophagus taken 2cm proximal to Barrett's oesophagus, we identified 3 which contained OSGs and in these there is clearly an increased number of cells expressing high levels of LEFTY1 and OLFM4 (see new **Figure 2c, Figure 5d and Supplementary Table 2**). Although the sequencing data need to be interpreted with care, and the detection of OLFM4/LEFTY1 populations was enriched by single cell sequencing methodology bias, the ability to detect OLFM4/LEFTY1 expressing OSG and BO cells together with their ability to resist cell death induced by experimental stress conditions highlights their functional similarity. All these data support, although do not prove, the notion that the expansion of OSG progenitors may cause BO – the manuscript discussion has been updated to reflect this.

5. Another important point is that it has been noted that the ESGC comprise of different cell lineages, including cells that carry markers for squamous lineages. These squamous cells were even found to possibly repopulate and give rise to neo-squamous epithelium after ablation and endoscopic mucosal resections. The authors do not point to at all to these different lineages nor to these papers. By IHC the expression of the squamous markers should be clearly noticeable in these glands. The authors do not point to such expression for instance in figure 1E. To my opinion the

only way to define the subpopulation of cells in the ESGC or submucosal glands is by dissection these glands from oesophageal specimens and subjecting these for single cell analysis, without preselecting cells for Epcam. This will indicate the different cell lineages and from which lineage the Barrett cells may arise.

We do agree with this reviewer that the best way to “define the subpopulation of cells in ESGC or submucosal glands is by dissecting these glands from oesophageal specimens and subjecting these for single cell analysis”. However to carry out the experiment suggested by this reviewer requires large pieces of oesophagus tissue from “normal”, non-cancerous patients and this is not permitted by our ethics. It is also practically impossible due to the difficulties of obtaining truly normal human oesophagus samples to collect sufficient numbers of OSGs for single cell sequencing. Additionally, as shown in our panel of 140 squamous endoscopic biopsies derived from 80 patients, the ability to detect OSGs from endoscopic biopsies is limited (only 3/140) (**Supplementary Table 2**). Therefore it is unfortunately not possible to follow the reviewer’s suggestion, as the number of endoscopic biopsies required and the cost of sequencing would be prohibitive, and is beyond the scope of this manuscript.

We understand why this reviewer suggested such an experiment, and part of this is due to their concern that the cells used in this study were preselected by EpCAM expression. However, as we explained in detail above, the cells used in this study were not selected by EpCAM expression, which we believe should allay the reviewer’s worries.

To address the question of whether cells with different squamous cell lineages are detected in OSGs, we used the identified glands derived from 2 of the endoscopic samples and one additional oesophagectomy sample, and detected KRT14-expressing cells in OSG ducts (please see updated **Figure 1e** and new **Figure 4c** K14/K7/MUC2 triple-immunofluorescent staining in OSGs under squamous epithelium 2cm away from the proximal Barrett’s border), consistent with published literature demonstrating they are bona fide OSGs. We subsequently used the adjacent sections from the same set of OSG-containing biopsies to examine the expression patterns of TFF3, LEFTY and OLMF4 and compared that to BO samples. Importantly we observed TFF3, LEFTY and OLMF4 expression in both OSG and BO samples examined (new **Figure 1e**, **Figure 2b and c**, and **Supplementary Table 2**). Together, our new validation data together with our existing single cell sequencing data have allowed us to reveal profound similarity between Barrett’s oesophagus cells and OSG cells. All these data support, although not prove, the hypothesis that Barrett’s may originate from OSG cells – the manuscript has been updated to reflect this.

6. The authors found LEFTY1 to be highly expressed in a large proportion of Barrett cells. Given the selection method of the cells this finding needs to be interpreted with care.

We agree with the reviewer that indeed, it is necessary to interpret these data with care. As the cell selection process was consistent across tissues, this finding is likely to reflect actual cellular composition of each tissue, as bias is consistent; however caution is necessary as there clearly are some innate differences between each tissue, as evidenced by different EpCAM expression in sorted cells between tissues.

During revision of this manuscript we have carried out extensive analysis of LEFTY1 expression in many different tissues (examples in **Extended Data Figure 2b**), and we observed that LEFTY1 is expressed throughout many tissues in the luminal GI tract. In non-diseased tissues, LEFTY1 positive cells appear to be isolated to deep gland structures; in a few cells in the colon, Brunner's glands, deep in pylorus and fundic crypts and in a few cells in OSGs under squamous oesophagus in patients without Barrett's. Importantly, from our analysis of 140 squamous biopsies and 140 Barrett's biopsies from 80 patients, we observed that LEFTY1 expression is high in OSGs and Barrett's, along with deep structures in gastric glandular epithelium and Brunner's glands of the duodenum. The staining patterns show that LEFTY1 expression is not restrained to the deeper parts of BO glands, which would explain why *LEFTY1* expression is differentially expressed in our analysis of BO cells from endoscopically obtained samples, which are generally superficial samples. Importantly, the ability to detect LEFTY1 expression in OSGs and in BO *in vivo* further supports our finding that the observed profound transcriptional similarity between OSG and BO cells does exist *in vivo* and is likely to reflect true biological similarities of these two cell types. It is not simply a result of an experimental artefact caused by the single cell sequencing methodology used in the study.

7. The authors only briefly mention the role of LEFTY1 and its association as a TGF-beta family member. Although no extensive data has been published on LEFTY1 and Barrett's, there are several papers that indicate the role of several TGF-beta family members, for instance in the pathogenesis of Barrett's. Also, there are several papers mentioning the regulation of LEFTY1 by other TGF-beta family members. It would be of high interest to perform pathway analysis in order to better understand and demonstrate the functional role of LEFTY1 in signalling. The RNA-seq data should easily allow such analysis.

We performed pathway analysis between the cells of each tissue. This has shown dysregulation of a large number of developmental pathways in the BO cells (**Review Figure**). However, we have not explored this pathway data further. Since it does not add to the conclusions we have drawn, we have not included it in the manuscript.

Review Figure: Gene ontology and pathway enrichment analysis in duodenum, gastric, BO and oesophagus cells

(a) Bar plots showing the p values of the top ten most dysregulated gene ontologies determined using goseq in R. Differentially expressed genes were generated by comparing cells from one tissue against all others (patients A-D, all with BO). **(b)** Bar plots showing the p values of the top ten most

dysregulated gene pathways curated by the Kyoto Encyclopedia of Genes and Genomes (KEGG). Differentially expressed genes were used as in a.

8. Minor points:

Abstract

-The authors mention 'acid' as the sole component in the refluxate of Barrett patients. This is incomplete since both bile and acids can be found in the refluxates of Barrett patients. The role of bile in the pathogenesis of Barrett's has been described in several papers. This needs to be corrected.

The abstract has been amended to address this point.

-correct the number of cells or do not mention a number that is overstating.

The abstract and text have been updated to address this point. Also, **Extended Data Figure 11** has been updated to summarise and clarify the cell numbers sequenced and analysed for each patient across all tissues.

-remove the word uncharacterized, since Epcam was used to sort the cells.

The text has been amended for clarity and **Extended Data Figure 10** provides additional detail regarding EpCAM expression and the effect of other cell sorting metrics on cell selection and sequencing performance.

Introduction

Correct the statement 'highly fatal'. Give proper numbers for survival and a reference. Do not refer to review articles but to original papers (e.g., ref 9)

References have been updated to address this and 'highly fatal' changed to 15% 5 year survival.

Result section

- It is mentioned that the normal patients had esophagitis. Was this also the case while taking biopsies?? If so this could have influenced the expression profiles of these cells, which needs to be addressed.

The 'normal' patients sampled did not have oesophagitis, these patients were attending endoscopy due to symptoms of reflux with PPI treatment but had no pathological findings on endoscopy or histology. This point has been clarified in the text.

- Include how many cells per biopsy/patients were finally included in the analysis.

We have updated **Extended Data Figure 11** to provide the necessary detail. We have also made changes in the text to include more detail (described above in the first response). We cannot

comment on cell numbers per biopsy in each individual case as for each tissue site, four biopsies were obtained in all cases and pooled.

- On page 7, ESGC are mentioned as rare cells. These glands are present in the proximal and distal esophagus and are functional (not 'remnants') in secreting mucins to lubricate and protect the esophageal mucosa. This needs to be corrected in the statements the authors made in the discussion. The glands are only rare in biopsies, because of their submucosal location. This needs to be corrected, also in the discussion.

We have corrected our statements in both the introduction and discussion.

- Figure 2: the LEFTY1 staining is not convincing. Is the expression cytoplasmic or nuclear? Better stainings are required and if still not clear I would encourage RNA-ISH to better identify the expression and include overview of the stainings.

We have carried out further LEFTY1 staining on an additional 140 squamous biopsies, 140 Barrett's biopsies, 142 Gastric biopsies and 135 duodenal biopsies from a cohort of 80 patients (please see **Methods** and new **Supplementary Table 2**). This has significantly improved our staining and we have replaced all immunohistochemistry images with improved quality images.

- Page 8. The authors state that 19% were (MUC2 positive) goblet cells. Did all patients had a number of goblets cells? Since in the material section it is stated that only patients with intestinal type of Barrett metaplasia were included, one would expect that all patients had a fraction of goblet cells.

We have therefore clarified the results with the statement "Since all patients had intestinal metaplasia, goblet cells made up 22%, 2.9%, 29% and 7.1% of cells in patient A-D, respectively." This data is visually represented in **Extended Data Figure 3**, however it is not possible to interpret this specific point from the figure. To aid in further cell identification, we have also included a metadata table in supplemental materials which details which cell belongs to which cluster in all analyses (**Supplementary Material 3**).

- Figure 1 represents an oesophagus with an adenocarcinoma in which submucosal glands are presumed to be positive for TFF3. However, in this case there is also an adenocarcinoma expressing TFF3. To me the structures pointed out as ESGC could as well be adenocarcinoma cell clusters. Specimens that better represent the ESGC should be included.

We now have this material in three patients with Siewart 2/3 tumours and no evidence of disease in the true anatomical oesophagus to replace the oesophagectomy specimen previously shown in **Figure 1**. The new data is shown in **Figure 1f**.

Extended Data Figure 11. Sample analysis and quality control

(a) Bar plot showing total number of cells sequenced of each tissue type by patient (A-D and G-H are patients with BO, E-F are patients without BO). Cells passing threshold for inclusion determined by controls (see **Methods**) are in solid colour, excluded cells are shown in faded colour. (b) Violin plots showing number of mapped reads by each tissue type, coloured as in a, in all cells which were sequenced. (c) Plot of number of genes detected (at least one read per gene) against total mapped reads for every sequenced cell and control. (d) Scatter plot showing the correlation between bulk tissue RNA-seq gene counts and an ensemble of single cell gene counts, taken from Barrett's samples from patient A, with Pearson correlation coefficient displayed.

a**EPCAM IMMUNOHISTOCHEMISTRY****b****c****d**
Extended Data Figure 10. Fluorescence activated cell sorting shows characteristics of cells selected for single cell RNA-sequencing.

(a) Immunohistochemical staining of EpCAM in oesophageal, Barrett's, gastric and duodenal epithelium. Scale bar 100 μ m. **(b)** Scatter plots showing the expression levels of the white cell marker CD45, the epithelial marker EpCAM, and the cell viability marker DAPI (high expression indicates non-viable cell) in all fluorescence activated cell sorting events, separated by cells which were sorted into 96 well plates (red), and those which were not (light grey). **(c)** Jitter plot showing the expression levels of EpCAM in plated duodenum, gastric, BO and oesophagus cells (coloured red in **b**) which passed the threshold for further analysis with those that did not. **(d)** Logistic regression comparing plated cells (coloured red in **b**) which passed the threshold for further analysis, set with positive and negative controls, with those that did not. Vertical black dotted line indicates p value threshold of 0.01, which none of the predictors exceed.

Extended Data Figure 5. Columnar gene-expressing cell detection in patients with and without Barrett's oesophagus

(a) Sankey diagram showing how oesophageal cells from patients with BO (A, B, D; all the cells in the oesophageal biopsies from patient C failed quality control and were not suitable

for analysis) and without (E and F) contribute to SC3 clusters generated using oesophageal cells only. **(b)** Jitter plots showing the expression of columnar (*TFF3* and *KRT7*) and squamous (*KRT14* and *TP63*) genes in cells from **a** coloured by patient (log10tpm) across each cluster. **(c)** Scatter plots showing the expression of *KRT7*, *KRT14* and *TP63* in the two clusters which contain the most *TFF3*-expressing cells (units are log10tpm). **(d)** Immunohistochemical staining of TFF3 in OSGs from normal oesophageal samples obtained endoscopically from two Barrett's patients (scale bars 400µm), with enlarged images (scale bars 50µm). **(e)** Immunohistochemical staining of TFF3 in OSGs under normal oesophagus taken from the proximal part of an oesophagectomy specimen resected for Siewert type III junctional tumour in a patient with no BO. Scale bars are 1000µm and 100µm in enlarged images.

Figure 1. Single cell RNA sequencing identifies cell groups in normal upper gastrointestinal epithelia

(a) Endoscopic sampling sites (yellow, oesophagus; green, gastric cardia; purple, duodenum; orange, Barrett's oesophagus) with summary of how tissues from patients were used. 2-4 biopsies were taken at each site. Patients without BO were sampled from the lower oesophagus 20mm proximal to the squamous-columnar junction. **(b)** From bulk RNA-seq data derived from samples from 13 patients with BO, heatmap of genes differentially expressed between any tissue type ($FDR < 1e-12$) with tissue hierarchy determined by nearest neighbour. Tissue indicated by colours as in **a**. **(c)** From bulk RNA-seq data, heatmap of expression of mucin and trefoil factor genes with tissue hierarchy determined by nearest neighbour, in samples from 13 patients with BO. **(d)** Upper panels show the cluster consensus matrices for single cells from normal tissue sites in four BO patients. Blue-to-red colours denote the frequency with which cells are grouped together in 250 repeat clusterings of simulated technical replicates (see Methods). Cell clusters are indicated by coloured bars below the matrices. In lower panels, heatmaps show expression of known functionally relevant genes that were differentially expressed between cell clusters (>4 fold change, $FDR < 1e-5$). **(e)** Haematoxylin and eosin staining of normal oesophagus taken from the proximal part of an oesophagectomy specimen resected for Siewert type III junctional tumour in a patient with no BO, showing OSGs (red arrow), OSG ducts (black arrow) and squamous epithelium (marked with dotted black line). Scale bar 500 μ m. **(f)** Immunohistochemical staining of KRT14, TFF3 and KRT7 (left, middle and right images, respectively) in adjacent sections from the same specimen as **e**, showing OSG ducts (black arrows) and OSGs (red arrows) and squamous epithelium (marked with dotted black line). Scale bar 500 μ m. OSG, oesophageal submucosal gland.

Figure 4. SPINK4 and ITLN1 mark early goblet cells

(a) Volcano plot showing fold change and p value of genes differentially expressed in the ‘goblet-type’ cell cluster as compared to all other cell clusters (see **Figure 3**). Points coloured

red indicate genes significant at 5% permutation test. Selected highly significant genes are labelled. **(b)** Bar chart showing the percentage of cells in the 'goblet-type' cell cluster (n=98) expressing *MUC2*, *ITLN1* or *SPINK4* alone or in different combinations (thresholds set at the tenth centile to include 90% of cells in which at least one transcript was detected from each gene). **(c)** Triple immunofluorescence staining images of MUC2 (red), ITLN1 (white) and SPINK4 (green) in normal colon from a resection specimen (blue stain is DAPI). Scale bar 100µm. **(d)** Triple immunofluorescence staining images of MUC2 (red), ITLN1 (white) and SPINK4 (green) in normal oesophageal epithelium obtained by endoscopic biopsy (blue stain is DAPI). OSGs encroaching on the surface epithelium are shown in the enlarged images on the right. Scale bars are 200µm and 50µm in enlarged images. **(e)** Triple immunofluorescence staining images of KRT14 (white), KRT7 (green) and MUC2 (red) in an OSG beneath normal squamous epithelium from an endoscopic biopsy of normal squamous epithelium from a patient with BO biopsy (blue stain is DAPI). Scale bar 50µm. **(f)** Representative immunofluorescence staining of Barrett's EMR specimen containing intestinal metaplasia but no dysplasia for MUC2 (red), ITLN1 (white) and SPINK4 (green); nuclei (DAPI) in blue. Scale bars are 400µm and 100µm in enlarged images.

Extended Data Figure 2. Expression of *LEFTY1*, *KRT7* and *KRT20* in BO, duodenal and gastric cells

(a) Jitter plots showing the expression of *LEFTY1*, *KRT7* and *KRT20* in Barrett's, duodenum and gastric cells. (b) Immunohistochemical staining of *LEFTY1* in normal tissues and Barrett's oesophagus as indicated. Scale bar 200µm.

Extended Data Figure 3. Clustering and differential gene expression profiles of all single cells from Barrett's oesophagus patient samples

(a) The left heatmap shows a cluster consensus matrix for single cells from all tissue sites in BO patients (n=1107 including brain positive controls). Blue-to-red colours denote the frequency with which cells are grouped together in 250 repeat clusterings of simulated technical replicates (see Methods). Tissue type is indicated below. Cell clusters are labelled on the right with the cell type they contain or a descriptive term if that cell type has not been

previously characterised. The right side heatmap shows differentially expressed genes (>4 fold, genes significant at 5% permutation test) in each cluster from panel **a**. Cells from each patient are indicated by the coloured bar on the left. **(b)** t-SNE plots of all cells from BO patients (n=1107 including positive controls), coloured in each panel by transcript level of a gene that is highly expressed in a particular cluster type (cluster name and gene name shown above plots).

Reviewer #2 (Remarks to the Author):

NB. Figures commented in this reply are attached at the bottom of the document in the order they are referred to in this reply (though not duplicated). All figures have been stylistically updated since the last submission. Additional figures to address specific reviewer points are inserted in-line in the response.

The main premise of this article is identification and characterisation of different cell types within Barrett's oesophagus (BE) and comparison of this tissue with other samples taken from GI tract. In order to study the samples, the authors used single cell RNA-seq, specifically the smart-seq2 methodology. Owen et. al. sequenced 2895 cells from 7 patients (5 patient with Barrett's oesophagus (BE) and 2 control patients) and identified a cell population within BE marked by LEFTY1 and OLFM4, which overlaps a subset of oesophageal cells. A hypothesis proposed by the authors is that this novel population of cells originates from the submucosal glands of normal oesophagus, and the authors postulate that the submucosal glands are the site of origin for BE.

This is an interesting hypothesis, investigated with a substantial sequencing effort.

There are a number of major comments that need to be addressed in a revised version of this manuscript:

1) The concept that Barrett's originates from submucosal gland cells is not new and although the single cell RNA seq data is elegant, no definitive evidence is provided to confirm or refute this hypothesis. A direct comparison with the single cell gene expression profile obtained from a squamous submucosal esophageal glands is essential to test this hypothesis.

We do agree with this reviewer that the best way to confirm the hypothesis that Barrett's originates from submucosal gland cells would be to carry out "a direct comparison with the single cell gene expression profile obtained from a squamous submucosal esophageal glands". However to carry out the experiment suggested by this reviewer requires large pieces of oesophageal tissue from "normal", non-cancerous patients, which is not permitted by our ethics. We also cannot carry out the required experiment with endoscopic biopsies as it is practically formidable and financially impossible. In our revised manuscript we have now shown that only 3 oesophageal endoscopic biopsies have histologically detectable OSG from a panel of 140 squamous endoscopic biopsies derived from 80 patients (full results in new **Supplementary Table 2**). Therefore it is impossible to address this reviewer's question as the number of endoscopic biopsies required and the cost of sequencing will be prohibitive and is beyond the scope of this manuscript.

Nonetheless, during the revision, we addressed this reviewer's question by carrying out immunohistochemical staining to validate the finding that OSG and BE cells express a similar set of genes using the identified three OSG-containing oesophagus biopsies derived from 2 endoscopic samples and one oesophagectomy sample. Consistent with published literature, the identified OSGs express detectable keratin 14 (KRT14) in gland structures, demonstrating they are bona fide OSGs

(please see revised new **Figure 1e-f indicated with black and red arrows**). We subsequently used the adjacent sections from the same OSG-containing biopsy and detected the expression of TFF3 and keratin 7 in cells located in the OSG. These results support the finding that OSG and BO cells share similar transcript profile.

Using the newly identified OSG-containing oesophagus samples, we also confirmed the expression of LEFTY and OLMF4, two of the new BO genes identified in this study, in OSGs (**See new Figure 2c and 5e respectively**). Importantly the similar expression patterns of TFF3, LEFTY and OLMF4 are detected in both OSG and BO samples (**Extended Data figure 5d and e and Supplementary Table 2**). Our new validation data, together with our existing single cell sequencing data, have allowed us to confirm that there is a profound transcriptomic similarity between Barrett's esophagus and OSG cells. All these data support, although not prove, the hypothesis that Barrett may originate from OSG cells.

Specific issues are:

- The authors state that expression of TFF2 and TFF3 in samples from squamous esophagus are consistent with these cells being from esophageal gland complexes - and this is reasonable. However, these trefoil factors are widely expressed in other parts of the GI tract, and therefore this information is not on its own sufficient to determine whether or not Barrett's arises from submucosal gland ducts.

We apologise for the confusion. The conclusion of Barrett's and OSGs being most similar epithelial types is based on the whole transcriptomic data, not just based on TFF2/TFF3 expression. Indeed, TFF2 and TFF3 are widely expressed throughout the GI tract and in our new **Extended Data Figure 1** we now show TFF2/TFF3 expression in all 4 tissue types derived from 919 single cells sequenced. For oesophagus we only included 120 columnar epithelial cells, labelled as OSG.

Extended Data Figure 2 shows a wider characterisation panel which indicates the similarity between OSGs and BO cells across a range of gene markers. To further address this question, we have now stained OSG cells with KRT7, LEFTY1 and OLMF4 and our updated **Figures 1e and f, 2b and c, 5c-e and Supplementary Table 2** show that both OSG and BO express these proteins.

- The authors use a method of elimination to deduce that a subset of BE epithelial cells have a gene expression profile consistent with their being ESGCs (p9, Fig3c and suppl Table 2). The comparison between BE cells with an ESGC profile and the transcriptional relationship of cells from other tissues (p9, last para) is helpful but not definitive.

This needs to be clarified in the presentation of the manuscript.

We observed profound transcript overlap between Barrett's oesophagus and OSG cells at a single cell level. Enteroendocrine cells from different tissues cluster with each other, not their tissue of origin. Goblet cells also preferentially cluster with other goblet cells rather than with their derived tissue. These specialised, differentiated cells were therefore removed from the analysis to reduce confusion and to check these cells did not drive the association between BO and OSGs. This analysis

was carried out on the main body of epithelial cells from each tissue without specialised epithelial cells. The manuscript has been updated to reflect this.

2) It is unclear how many cells were included for each analysis. Furthermore, it is not entirely clear how cells from each patient contributed to the results. Please make this information transparent, since it is important for determining whether interpatient batch effects are contributing to the clustering. Based on our analysis of the supplementary data used for clustering in the Figure 1, it seems as though the conclusions reached depend on a small number of cells arising almost entirely from a single patient.

To clarify the number of cells used for sequencing and analysis, we have now included a new **Extended Data Figure 11** to clearly illustrate the numbers and types of cells derived from each patient. During the revision, we have used biopsies derived from further two patients to carry out additional single cell sequencing. Therefore we have used biopsies from eight patients in total (patients A-H). The text has now been modified to reflect the new information. “A total of 4237 cells were sequenced from 8 patients (Extended Data Table 1) in three batches. Due to known issues with batch effects in single cell experiments²⁵, analysis of cells from each batch has been kept separate where feasible and cells were permuted across plates and pooled prior to sequencing (see Methods). The first batch yielded 1040 cells (207 duodenum, 227 gastric, 371 BO and 235 oesophagus) suitable for analysis from four patients (A-D) with BO and intestinal metaplasia. A total of 214, 35, 66 and 56 BO cells were analysed from each BO patient, respectively. The second batch yielded 648 oesophagus cells suitable for analysis from two patients (E-F) with symptoms of gastro-oesophageal reflux but no identifiable oesophageal pathology. Finally, the third batch of cells yielded 194 cells (29 pylorus, 109 gastric, 32 BO and 24 oesophagus) suitable for analysis from two patients (G-H) with BO and intestinal metaplasia. Overall, a mean of 1.2×10^5 reads were mapped per cell and a median of 3978 genes were detected per cell (with at least one read per gene).”

Based on our understanding of the clusters, we conclude that:

- **The TFF2/TFF3 high cells derive almost entirely from Patient A.**

We agree with the reviewer’s observation that most *TFF3* high cells are derived from patient A. We have now analysed oesophageal cells from patients with and without BO in isolation and show that patient D also contributes many *TFF3* positive oesophageal cells (**Extended Data Figure 5a-b**). In fact, *TFF3*-expressing cells are also derived from patients B and D. Additionally, oesophageal cells from patient E, who does not have identifiable Barrett’s oesophagus, express *TFF3*, suggesting *TFF3* cells may exist in normal OSGs. *TFF3*-expressing cells cluster into clusters 3 and 5. Cluster 5 consists of cells from patient A and patient B. Cluster 3 consists of cells from Patients B, D and E. Patients A, B and D had Barrett’s oesophagus. Patients E and F had no evidence of Barrett’s oesophagus, but had previously reported reflux symptoms).

During the revision, we also identified three further tissue samples with OSGs to carry out immunohistochemical staining, and we observed similar expression patterns of *TFF3* as well as our newly identified BO genes such as *LEFTY1* and *OLFM4* in both OSGs and BO sections (**new Figure 1f**,

Figure 5e and Extended Data Figure 5), supporting the finding that the cells we identify in our single cell RNA-seq data are representative of 'typical' OSGs.

- **For the esophageal data patients B and D have almost no cells in cluster 1. Patient C had no esophageal data. One would expect them all to have expression confirming the esophageal origin.**

Oesophageal submucosal glands in an endoscopic biopsy is an uncommon event, as we now show in the analysis of 140 samples from 80 patients (as described in the first response). Obtaining substantial quantities of OSG cells in each single cell experiment is unfeasible using our approach. All the cells in the oesophageal biopsies from patient C failed quality control and were not suitable for analysis. As an alternative, we have now identified 3 additional OSG samples in normal squamous epithelium and have characterised these further using an immunohistochemistry study to support our claims.

- **Data for patient A indicates that around 50% of the esophageal samples (either within a single biopsy or 50% of individual biopsies) fall into the TFF2/TFF3-high group. Presence of such a large population of cells should be observed within the bulk RNA-seq data. This is not observed in figure 2c**

The high percentage TFF- expressing cells (50%) detected in the oesophageal sample from patient A is unusual. As shown in our new **Extended Data Figure 5** and in our new validation study shown in the new **Figure 1**, our ability to detect high TFF3-expressing OSG cells from endoscopic oesophageal biopsies is a rare event (3/140 oesophageal endoscopic biopsies). This is the reason why TFF3 expression is hardly detected in the oesophageal biopsies bulk RNA-seq data (**Figure 1c**, please note additional data added as new patients were sequenced)

- **The gastric profile varies enormously between patients. The duodenal samples have a much better representation across the clusters suggesting that the data are comparable.**

We are not clear on how this profile varies between patients in the context of the other cell types examined. However one possible explanation is the fact that gastric tissue is most exposed to gastric acid and this may affect cells' viability. Consistent with this, the number of gastric cells to pass the required criteria for single cell analysis is generally the lowest compared to other tissues. During the revision, we attempted to sequence more cells from gastric and include cells from the gastric pylorus, however the above mentioned effect is even more marked in these poorer quality data and especially so in the pylorus (**Extended Data Figure 11**). This effect will be interesting to investigate with higher throughput single cell sequencing techniques in the future. Nonetheless, for the gastric cells that passed the quality check, most gastric cells sit in a mucus neck/foveolar cell cluster as expected, with some enteroendocrine cells seen from each patient suggesting that some cells can survive the acid condition and the experimental conditions required for single cell sequencing to provide us with meaningful transcriptomic profiles.

- **As a quality check, it might be beneficial to compare the bulk RNA-seq from patient A with the**

combined scRNA-seq data for that patient, in order to confirm that the average scRNA-seq signal is similar to bulk RNA-seq

Following this reviewer's request, we have now compared bulk RNA-seq and combined single cell RNA-seq for the BO sample from patient A (as this has the highest number of cells representing a single tissue and patient). The correlation between these two RNA-seq data is high (0.89) and this data is now shown in **Extended Data Figure 11d**.

3) The markers of "early goblet cells" are correlative and not conclusive. Functional studies would be required to take these cell lineages and show that they can differentiate into goblet cells.

To investigate whether ITLN1 and SPINK4 marks early goblet cells, we grew human duodenal organoids from 3 patients endoscopic biopsies (**Review Figure 1**). We treated the organoids with a gamma secretase inhibitor at 0, 24 and 48 hours to induce differentiation of secretory lineages (see below). We observed that untreated organoids had a higher level of SPINK4 expression (green) compared to organoids at 48 hours. Interestingly, there is an inverse association between MUC2- (red) and SPINK4-expressing cells in the same panel of organoids. ITLN1 staining (white) is seen at low levels throughout the organoids at all stages of treatment. This expression pattern is in agreement with the *in vivo* expression patterns of SPINK4, MUC2 and ITLN1 in colonic crypt. In particular, SPINK4-expressing cells tend to locate at the lower part of the crypt, which is suggestive of early differentiating secretory cells whereas MUC2-expressing cells tend to locate at the top of the crypt, indicative of secretory cells lineages later in the differentiation process. Importantly this expression patterns of SPINK4, MUC2 and ITLN1 is not observed in Barrett's oesophagus glands with intestinal metaplasia. All these suggest that in the normal crypt and during differentiation of secretory cell lineages, SPINK4 and ITLN1 may mark an earlier cell lineage than that of MUC2. The same staining pattern has been reproduced 3 times from organoids derived from three different patients. However the technical difficulties of triple staining with 3 rabbit antibodies with effective matrigel penetration has impaired the ability to produce clearer staining. Also, although organoids are a widely used and useful model, they do not allow for a growth factor gradient enabling gradual differentiation up the gland. Hence the more widespread staining pattern observed in these organoids. Future technical improvement is needed to establish an ex-vivo culture condition that reflect *in vivo* differentiation status closely and to be able to generate more clear images than our current conditions. Therefore the organoid result is only included for this reviewer but they will not be included in the revised manuscript because a large amount of work will be needed to address this question properly and it is beyond the scope of this manuscript.

Review Figure 1. Organoid cell differentiation model showing interval expression of MUC2, ITLN1 and SPINK4.

Duodenal organoids grown from endoscopic biopsies treated with gamma secretase inhibitor for 0hrs, 24 hrs and 48 hours. Immunofluorescence stained for ITLN1 (white), MUC2 (red), SPINK4 (green) and DAPI (blue). Scale bar 50um.

4) The authors sought a stem-like population in the BE and ESGC cells by applying an algorithm to the single cell RNA profile. The highest scoring cluster (C3 in Figure 5b) was predicted as a stem-like population, however it is not shown whether this stem-like population is found in both BE and ESGC cells, or in all patients. It raises the same issue as the aforementioned ESGC cells, and appears to be another n=1 case.

The contributions each patient and tissue have made to this 'high scoring cluster' are now presented in an additional plot in **Extended Data Figure 8b**. We have also updated the text to address this with the following statement: "Cells from all four patients with BO contributed to this cluster, including oesophageal cells from two patients". Furthermore, we identified 3 squamous biopsies containing OSGs which were all also positive for OLFM4. Interestingly, these OSGs obtained from patients with BO have more OLFM4 expression than in OSGs from non-Barrett's patients. OLFM4 staining can be seen throughout all Barrett's glands in 140 endoscopic biopsies from 80 patients (**Supplementary Table 2** and **Figure 5**), this diffuse pattern is not seen in the gastric epithelia or duodenum (**Extended Data Figure 8d**)

5) The clinical implications of the findings are over-stated. Barrett's is a straightforward diagnosis made from endoscopy and histopathology, and molecular analysis is not required, and indeed would add considerable financial and logistical burden. The value of molecular analyses maybe to help differentiate those patients at high risk for cancer progression, but this is not the topic of this manuscript.

We have amended the text to address this comment by removing the suggestion that further molecular analysis would help with the diagnosis of BO, as this is straightforward.

6) The authors state that this study does not provide data relevant to cancer progression in Barrett's and that further studies will be required to inform on this aspect. However, 8/19 samples were taken from BE with dysplastic foci (Extended Table 2). Therefore, it is not clear whether some of the transcriptional changes do in fact reflect the progress towards cancer.

Although we agree with this reviewer that it will be important for us to examine whether some of the observed transcript changes could reflect the progress towards cancer, our current data are under-powered to address this. None of the transcript data originate from biopsies from patients with dysplasia. All biopsies which were sequenced had frozen sections taken, were analysed by two consultant histopathologists, seemed free of dysplasia and had the presence of goblet cells before sequencing. Staining validation of transcript markers have been validated using immunohistochemistry on 140 biopsies with non-dysplastic Barrett's oesophagus (**Supplementary Table 2**). Similar expression patterns of identified BO genes such as TFF3, LEFTY1 and OLFM4 were observed in BO biopsies and EMR specimens with or without dysplasia. The patients listed in Supplementary Table 2 are not the same as the patients listed in Extended Data Table 1 (the sequenced cohort), therefore we do not have sufficient numbers of dysplastic or malignant samples to evaluate whether some of the identified new BO genes expression patterns (LEFTY1 and OLFM4 for example) may reflect BO to OAC conversion.

7) The introduction states that the current most widely held view is that Barrett's originates from the stomach. This has been a controversial topic undergoing research for over 20 years with a number of competing theories, and it is not clear from the publications that there is a current consensus. The theory that Barrett's originates from submucosal glands of the squamous esophagus is not new; however the authors have neglected to cite the previous research about the submucosal gland duct as a possible source of origin for Barrett's. The previous data should be mentioned (see recent review on Origin of Barrett's epithelium in Cellular and Molecular Gastroenterology (Ref 1 below) and Hepatology (Ref 2 below) published which summarises much of the published evidence).

Thank you for the reviews. We have updated the text and have referenced some of the papers discussed to provide a more balanced perspective.

8) As noted in the introduction of the manuscript, a number of gastric sites have features

recapitulated in Barrett's epithelium, and indeed the title of ref 11 is "gene expression profile of Barrett's epithelium replicates pyloric-type gastric glands". Therefore, the authors need to justify why they only sampled the gastric cardia region for comparison. In line with the hypothesis to "clarify the relationships between cells in normal tissues and BE" a wider upper GI sampling protocol would have allowed more robust conclusions to be drawn. This needs to be clarified in the presentation of the manuscript.

To address this reviewer's comment, we carried out single cell sequencing using endoscopic biopsies taken from oesophagus, BO, gastric cardia and pylorus derived from two additional BO patients (patients G and H in **Extended Data Figure 11d**). Due to sampling limitations imposed by our ethics, we replaced the duodenal samples with pyloric samples rather than simply adding a further sampling site. Whilst this additional data is of much poorer quality than the original data set, it does demonstrate that there are few transcript differences between pylorus and cardia cells (**Extended Data Figure 7**).

Additionally, we also carried out immunohistochemical staining of TFF3, LEFTY1 and OLFM4 in 140 proximal gastric biopsies from 80 patients and 3 gastric pylorus samples and compared their expression pattern with that observed in 140 BO biopsies derived from the same set of 80 patients. It is important to note that TFF3 and LEFTY1 staining is higher in BO and OSGs than any other tissue, although their expression is seen deep in gastric glands and in Brunner's glands (**Extended Data Figure 2**). OLFM4 is also seen in all tissues. The staining pattern is diffuse in BO and OSG, but isolated to sparse individual cells in gastric epithelium and Brunner's glands. OLFM4 is seen throughout the duodenal crypt base (**Extended Data Figure 8**)

9) Please can the author clarify from where normal squamous oesophageal biopsies were taken from? This is important, as the density of submucosal glands varies according to the proximal-distal extent of the oesophagus. It appears that squamous samples showing a submucosal gland profile were taken from above the Barrett's- depending on the level in that patients the density of submucosal glands might be expected to be very low. This would also beg the question as to why the Barrett's did not extend higher if a submucosal gland was present at this location.

Sampling was consistent with samples taken 20mm proximal to the Barrett's lesion (most proximal extent) or 20mm proximal to the SCJ in patients with no Barrett's. The manuscript has been updated to clarify this. Future studies are needed to map the exact density of glands in a proximal-distal fashion and this is beyond the scope of this manuscript.

10) As above, clarification is needed to understand which samples were used for which experiment. When comparisons were made between transcriptional profiles of Barrett's and squamous samples was this between squamous epithelium of control patients only or Barrett's and controls? For Barrett's patients the normal squamous was taken from the proximal oesophagus above the Barrett's, and presumably in controls the squamous sampling was performed closer to the gastro-oesophageal junction.

The sampling protocol was kept consistent as described above. The new **Extended Data Figure 11** helps to clarify the number of cells from which tissue and which patient were used.

The control patients (non-Barrett's patients) were sampled and processed at a different time point, thus, to avoid a sequencing batch effect, the analysis with Barrett's cells was kept separate. In the case of **Extended Data Figure 5**, the BO cells have now been removed and this analysis is only applied to OSG cells from patients (A-F).

11) Hierarchical clustering showed that Barrett's samples were more closely related to gastric tissues. Was any attempt made to quantify the amount of IM present, since focal versus diffuse intestinalisation will likely affect this result?

The expression of MUC2 can be seen in the whole tissue samples in **Figure 1c**. This shows that Barrett's samples with lower MUC2 expression levels did cluster more closely with some gastric samples, and that the higher MUC2-expressing Barrett's samples clustered together more closely. Following this reviewer's comment, we analysed the expression levels of intestinal metaplasia-specific genes in the sequenced BO single cells, which confirms the presence of MUC2 positive cells in all patients (**Review Figure 2**). Whether this correlated with the histological findings is difficult to say on the frozen sections we obtained. Two pathologists individually scored slides as Barrett's glands with intestinal metaplasia; however no quantification is possible histologically as we no longer have any more tissue to analyse from these patients. However, when cells are clustered across all tissues (**Figure 3**), there is a strong relationship between goblet cells (and enteroendocrine cells, the other main group of specialised cells we identify) irrespective of their source. These effects will drive the hierarchical clustering seen in bulk RNA-seq, whereas our single cell analysis confirms that many gastric cells (mucus-neck type) clustered independently, and that BO non-specialist cells clustered with the columnar gene-expressing oesophageal cells.

Review Figure 2. MUC2 expression in cells from BO patients

A box plot showing the expression of MUC2 (log10 tpm) in all cells analysed from each of the four patients (A-D) with Barrett's oesophagus.

12) Which other developmental genes were expressed in Barrett's versus other tissues? For example, HOX gene profiles have been interrogated in Barrett's previously with some evidence that the profile might most closely resemble HOX expression patterns observed in colonic tissues. In the Supplementary Table 2 it seems that HOXB6 is detected - is this correct? An explanation would be helpful.

This reviewer is correct that HOXB6 was identified as a gene upregulated in Barrett's (Supplementary Table 2) and, in agreement with previous findings, increased HOXB6 expression in BO was also observed in bulk sequencing. As the aim of this paper is to identify previously unrecognised BO genes that are highly expressed at a single cell level but not at the bulk RNA-seq level, we therefore focused on genes such as LEFTY1 and OLFM4. Nonetheless, we agree with this reviewer that it is important to acknowledge these findings, hence pathway analysis was completed between cells from each tissue (**Review Figure 3**). This has shown dysregulation of a large number of developmental pathways in the BO cells. However, we have not explored this pathway data further and thus have not included it in the manuscript.

Review Figure 3. Gene ontology and pathway enrichment analysis in duodenum, gastric, BO and oesophagus cells

(a) Bar plots showing the p values of the top ten most dysregulated gene ontologies determined using goseq in R. Differentially expressed genes were generated by comparing cells from one tissue against all others (patients A-D, all with BO). **(b)** Bar plots showing the p values of the top ten most

dysregulated gene pathways curated by the Kyoto Encyclopedia of Genes and Genomes (KEGG). Differentially expressed genes were used as in a.

13) It is not clear to me why the single cell RNAseq data on p7. shows that “cell subpopulations, including rare subpopulations from ESGCs that cannot be identified by conventional RNA-seq”. We have known for years from H&E analysis of paraffin sections about submucosal glands in the human distal esophagus. The acute phase response genes are interesting but the significance of these findings is not clear.

We agree the acute phase response genes are interesting. Further studies need to be done to characterise the role of acute phase response genes. The new **Extended Data Figure 5a and 5b** are good examples of why single cell RNA-sequencing is able to identify rare cell subpopulations whereas conventional bulk RNA-seq cannot. For example, high TFF3-expressing cells were detected from oesophagus biopsies derived from patients A, B, D, E. However for many patients, (B, D, E), only a few high TFF3-expressing cells were present. This cell population cannot be identified by conventional bulk RNA-seq, and consistent with this finding, TFF3 expression is barely detectable in corresponding bulk oesophagus RNA-seq data. Finally, and consistent with our finding, the rarity of detecting OSG is confirmed in our validation study where only 3/140 endoscopic oesophagus samples had detectable OSGs.

One advantage of single cell sequencing is to allow the identification of cell populations that cannot be teased out from bulk RNA-sequencing, therefore allowing creation of a cell subpopulation ‘pseudo-biopsy’ from a tissue or lesion. Whilst some reads undoubtedly map to columnar genes in bulk sequencing of squamous samples containing OSGs, the polluting effect of squamous material prevents a focussed examination of their expression. The recognition of acute phase response was merely to highlight why the squamous-characterised oesophagus cells fall into two clusters in the analysis in **Figure 1d**. Since we have not made any specific conclusions based on this observation, this was not discussed further.

14) If MUC2 marks goblet cells why were these present in the gastric samples (p10, Fig 4)? Whether ITLN1 and SPINK4 mark early goblet cells is an interesting idea, but it is not proven by this correlative analysis.

In the results shown in Figure 4a, we did detect a single gastric cell expressing MUC2. The reason for this unexpected finding is currently unknown since MUC2 expression was very low in gastric cells (**Extended Data Figure 1**). Whether this is a contamination from duodenal sample or a sample swap, or represents a true example of gastric IM in this patient (but not seen in histology) is impossible to answer. However it does not change the findings or interpretation. To avoid confusion, we have changed **Figure 4a** to a volcano plot, consistent with the presentation of RNA-seq data.

To investigate whether ITLN1 and SPINK4 mark early goblet cells, we used human duodenal organoids treated with a gamma secretase inhibitor for up to 48 hours to induce differentiation of secretory lineages. We observed that in an untreated organoid or the one treated for 24 hours, more SPINK4-expressing cells (green) were detected than that in 48 hours treated organoid. Interestingly

there is an inverse association between MUC2- (red) and SPINK4-expressing cells in the same panel of organoids. ITLN1 staining (white) is seen at low levels throughout the organoids at all stages of treatment. This expression pattern is in agreement with the *in vivo* expression patterns of SPINK4, MUC2 and ITLN1 in the colonic crypt. In particular, SPINK4-expressing cells tend to locate at the lower part of the crypt suggestive of early differentiating secretory cells whereas MUC2-expressing cells tend to locate at the top of the crypt, indicative of secretory cells later in the differentiation process. Importantly, this expression patterns of SPINK4, MUC2 and ITLN4 is not observed in Barrett's oesophagus glands with intestinal metaplasia. The same staining pattern has been reproduced 3 times from organoids derived from three different patients. However the technical difficulties of triple staining with 3 rabbit antibodies with effective matrigel penetration has impaired the ability to produce clearer staining. Also, although organoids are a widely used and useful model, they do not allow for a growth factor gradient enabling gradual differentiation up the gland. Hence the more widespread staining pattern observed in these organoids. Future technical improvement is needed to establish an *ex-vivo* culture condition that reflect *in vivo* differentiation status closely and to be able to generate more clear images than our current conditions. Therefore the organoid result is only included for this reviewer but they will not be included in the revised manuscript because a large amount of work will be needed to address this question properly and it is beyond the scope of this manuscript.

15) The volcano plot in Figure 5c (below) needs clarification. OLFM4 is selected as the stem-like cell marker because of its high expression in the highest-scored cluster C3 and its association with classic stem cell marker LGR5. However, the plot shows 5 red dots, does it mean there are another 4 genes similarly expressed as OLFM4?

Using StemID algorithm, we did detect high CPS1, PIGR, ANPEP, A2M in addition to LGR5 and OLFM4 as potential stem-like cell markers. These results are available in full in Supplementary Table 4. However as the biological importance of many of the identified genes are unknown and this is the first single cell paper applying this methodology in Barrett's oesophagus, we were cautious with our finding and do not want to propose genes that have not previously been implicated in stemness in relevant tissues as new stem cell markers. Future studies are needed to investigate this further.

16) The faint staining of OLFM4 in ESGC looks non-specific (see also minor comment below). Because this study does not provide functional or clonal assays, which are normally required for stem cell science, it is important to show clear expression patterns of the markers in tissue, as well as co-expression of classic stem cell markers/associated genes.

We appreciate the staining was poor in this tissue. During the revision, we have carried out OLFM4 staining in 140 biopsies from 80 patients with Barrett's oesophagus and three OSG samples identified from the same 80 patients. We observed that OLFM4 staining is in a few cells per duct in patients without Barrett's. In 3/140 biopsies where OSGs were observed OLFM4 staining was more diffuse and seen throughout the whole duct (**Figure 5d** and **Supplementary Table 2**). In **Extended Data Figure 8d** OLFM4 is shown in a few individual cells in the gastric fundus, pylorus and Duodenal Brunner's glands. In 140 biopsies from 80 patients with Barrett's oesophagus we observed the same

diffuse cytoplasmic staining in many cells throughout every Barrett's gland, suggesting expansion of the OLFM4 cell population.

17) The study then focused on the co-expression pattern of OLFM4 and LEFTY1 in two subsets of BE cells i.e. B2 and B3 in Figure 2a. The representation of the stem-like population in ESGC cells from normal esophagus is not mentioned.

During the revision, we have carried out OLFM4 and LEFTY1 staining in all OSGs and BO samples (**Figure 2, Figure 5, Extended Data Figures 2, Extended Data Figures 8 and Supplementary Table 2**). We observed that OLFM4 stains a discrete cell population in the OSGs in patients without Barrett's. Interestingly there is expansion of this cell population in patients with Barrett's oesophagus, even in OSGs derived from the oesophageal epithelium proximal to the Barrett's. Future studies are needed to investigate the significance of these expression patterns.

18) The subtitle: "OLFM4 marks stem-like transcriptional behaviour in columnar esophageal epithelium" (Line 221) is over-stated as the evidence mainly supports OLFM4 expressing stem-like subset residing in BE cells but not ESGC cells.

This result was drawn from combining RNA expression detected in BO and OSG cells. During the revision, we have now obtained new data detecting OLFM4 positive cells in both all OSGs and Barrett's oesophagus examined by immunohistochemistry, we have now changed the subtitle to "OLFM4 marks a stem-like transcript profile in BO and OSG epithelium".

19) Given that another widely held theory is that BE originates from gastric stem cells, it is a little odd that this study did not also use the StemID algorithm to analyse gastric cells.

To address this reviewer's comment, we carried out StemID of the gastric cells, in addition to Barrett's and OSGs in isolation (**Extended Data Figure 8a and b**, full results available in **Supplementary Table 4**). Unfortunately we did not detect any known gastric stem markers such as LGR5 in our gastric single cell data using StemID, however some expression of LGR5 was seen in Barrett's, duodenum, gastric and OSG cells (LGR5 expression has now been added to **Extended Data Figure 1**). This is consistent with our finding that large number of sequenced gastric cells are differentiated mucus neck/foveolar cell types. Interestingly, however, SMURF2 was identified in the high scoring cluster which is known to interact with activated TGF-beta-specific R-Smad, Smad2, and to induce its ubiquitination and degradation (Lin et al. Smurf2 Is a Ubiquitin E3 Ligase Mediating Proteasome-dependent Degradation of Smad2 in Transforming Growth Factor-β Signaling. JBC, 2000.) While a number of genes have been identified as stem-like genes by the StemID algorithm, we would not be happy to propose them as a new stem marker for gastric tissue as their biological role in gastric stem cells are unknown.

20) Extended data 8a: The H&E images are poor quality and show the surface of the epithelium and need higher power.

These H&E images are from 15micron frozen sections from the same biopsy that were subsequently used to carry out single cell RNA sequencing analysis. Since these are not 4 micron paraffin fixed tissue sections, their resolution is suboptimal. However we have now included images with 20x and 40x magnification although the resolution is still limited.

Minor comments:

- Abstract: Barrett's occurs in response to efflux of acid and bile, not just acid

We have updated the manuscript to address this.

- Results: The % of BE cells expressing LEFTY1 is given but no % for other genes listed. This should be added to enable comparisons to be made.

We have updated the manuscript to address this for MUC2. If there is value in adding more detail to control tissues, we are happy to do so, however this does not link to any discussion points so has been omitted for brevity in this resubmission.

- KRT7 expression is noted as being "consistent with its clinical utility in BE diagnosis". This is incorrect - KRT7 expression is not used clinically for Barrett's diagnosis.

We have updated the manuscript to address this. KRT7 is mentioned in the 2013 British Gastroenterology Guidelines, but not in a clinical utility context, hence our mistake.

- The origin of colonic samples should be stated in methods and ethical approvals.

We have updated the manuscript to address this. Colonic samples came from the same biobank approval as our BO samples.

- The ESGCs could be viewed as a developmental remnant, however they do have a physiological role in mucous production to protect the squamous epithelium.

We have updated the manuscript to address this and changed calling them remnants in view of the physiological role they provide.

- Fig. 4e: the BE glands look like epithelial glands rather than submucosal esophageal glands that can also sometimes be seen beneath Barrett's epithelium.

We have clarified the legend to describe what this figure is (now moved to **Figure 4f**).

'Representative immunofluorescence staining of Barrett's EMR specimen containing intestinal metaplasia but no dysplasia for MUC2 (red), ITLN1 (white) and SPINK4 (green); nuclei (DAPI) in blue.'

- **Fig. 5: Stem-lie should read stem-like. The OLFM4 staining in ESGD looks like nonspecific background staining (image 5f).**

We have now replaced the old images with new images using the identified OSG material (**Figure 5**). Spelling mistake has now been also corrected.

- **Extended data Table 2. The F:M ratio is surprising for this disease.**

Yes, these were taken on consecutive sampling sessions, hence the distribution in this small sample. With the addition of more data, this is more balanced than previously.

- **Methods. How did you define “mild reflux” for your control patients?**

“Mild reflux” refers to patients suffering from 0-2 acid reflux episodes per week along with normal endoscopic appearances of the upper gastrointestinal tract and no histological evidence of oesophagitis in processed samples. The manuscript has been updated to include this.

- **Ref 2 is incomplete and the link or detailed source should be given**

CRUK provides detailed cancer information for patients and regularly updated epidemiological details in the UK. This has now been updated to “CRUK. <http://www.cancerresearchuk.org/about-cancer/oesophageal-cancer>. (2016)”.

- **Supplementary tables and material require legends and detailed description of content**

This has been added.

- **The ordering of clusters in the supplementary material is different to the main figures**

The ordering of clusters in the Supplementary Materials and main figures have been matched.

- **High quality images are require for figures 1e, 2b, 4c-e, 5e,f, S8a**

Low quality images are caused by the journal submission restrictions. Higher quality images are available upon request. Also, most images have now been updated from the original submission.

- **Methods section refers to figure S6, we think it should actually be S8 (lines 525, 544, 571)**

Apologies, this has been corrected.

- **Line 297 – most of the article is written in American English, however ‘colours’ uses British spelling**

In view of the journal, all spelling has been changed to British, with exception of references or previously named software/algorithms.

- There is not information provided on whether raw data will become available

Raw data will be available. Unfortunately submission is complex due to anonymity requirements and our submission to the SRA was denied as we are Europe based. We have begun our submission to the European Genome-phenome Archive.

- Methods: Please include the concentration of ERCCs added to the single cell RNAseq samples.

The ERCC spike-in was added at a 1:100,000 dilution to input total RNA. This information has been added.

References

1. Garman, K. S. Origin of Barrett's Epithelium: Esophageal Submucosal Glands. *Cell. Mol. Gastroenterol. Hepatol.* 153–156 (2017). doi:10.1016/j.jcmgh.2017.01.016
2. van Nieuwenhove, Y., Destordeur, H. & Willems, G. Spatial distribution and cell kinetics of the glands in the human esophageal mucosa. *Eur. J. Morphol.* 39, 163–8 (2001).

Figure 1. Single cell RNA sequencing identifies cell groups in normal upper gastrointestinal epithelia

(a) Endoscopic sampling sites (yellow, oesophagus; green, gastric cardia; purple, duodenum; orange, Barrett's oesophagus) with summary of how tissues from patients were used. 2-4 biopsies were taken at each site. Patients without BO were sampled from the lower oesophagus 20mm proximal to the squamous-columnar junction. **(b)** From bulk RNA-seq data derived from samples from 13 patients with BO, heatmap of genes differentially expressed between any tissue type ($FDR < 1e-12$) with tissue hierarchy determined by nearest neighbour. Tissue indicated by colours as in **a**. **(c)** From bulk RNA-seq data, heatmap of expression of mucin and trefoil factor genes with tissue hierarchy determined by nearest neighbour, in samples from 13 patients with BO. **(d)** Upper panels show the cluster consensus matrices for single cells from normal tissue sites in four BO patients. Blue-to-red colours denote the frequency with which cells are grouped together in 250 repeat clusterings of simulated technical replicates (see Methods). Cell clusters are indicated by coloured bars below the matrices. In lower panels, heatmaps show expression of known functionally relevant genes that were differentially expressed between cell clusters (>4 fold change, $FDR < 1e-5$). **(e)** Haematoxylin and eosin staining of normal oesophagus taken from the proximal part of an oesophagectomy specimen resected for Siewert type III junctional tumour in a patient with no BO, showing OSGs (red arrow), OSG ducts (black arrow) and squamous epithelium (marked with dotted black line). Scale bar 500 μ m. **(f)** Immunohistochemical staining of KRT14, TFF3 and KRT7 (left, middle and right images, respectively) in adjacent sections from the same specimen as **e**, showing OSG ducts (black arrows) and OSGs (red arrows) and squamous epithelium (marked with dotted black line). Scale bar 500 μ m. OSG, oesophageal submucosal gland.

Figure 2. *LEFTY1* and *OLFM4* are mainly expressed in Barrett's oesophagus cells that do not express differentiated secretory cell markers

(a) Upper panel, cluster consensus matrix of BO cells from 4 BO patients (n=371 cells). Blue-to-red colours denote the frequency with which cells are grouped together in 250 repeat clusterings of simulated technical replicates (see Methods). Clusters (B1-B4) are indicated by the coloured bars below. Lower panel, heatmaps showing expression of selected functionally relevant genes that are differentially expressed between cell clusters (>4 fold change, FDR <1e-5). (b) Immunohistochemical staining of MUC2, LEFTY1 and CHGA in sections derived from the same BO resection specimen. Black arrows indicate goblet cells on all sections (positively stained for MUC2; negative for LEFTY1 and CHGA) Scale bars are 50µm. (c) Immunohistochemical staining of LEFTY1 in an OSG from a normal squamous endoscopic biopsy obtained from a patient with BO. Scale bars are 300µm and 50µm in enlarged image.

Figure 5. *OLFM4* is upregulated in BO and OSG cells with stem-like transcript profiles

(a) Bar plot on left shows StemID scores across all RaceID2 clusters (see **Methods**) applied to all non-squamous oesophageal cells (BO and oesophageal cells with <5 KRT14 counts to exclude squamous cells, n=533). Scores are calculated from multiplication of the entropy

(spread from the cluster mean) and the number of cluster links arising from a given cluster. Differentially expressed genes in the highest scoring cluster (C3, blue asterisk) and second highest scoring cluster (C7, red asterisk) are shown in the volcano plots in the centre and right plots, respectively. Points coloured red indicate the most significant genes with a fold change greater than 2. Selected highly significant genes are labelled. **(b)**

Immunohistochemical staining of OLFM4 in human colon (close-up of base of crypt inset). Scale bars are 100 μ m and 20 μ m in inset. **(c)** Immunohistochemical staining of OLFM4 in BO mucosal resection containing intestinal metaplasia but no dysplasia, with enlarged image. Scale bars are 1000 μ m, 200 μ m in enlarged image and 50 μ m in inset. **(d)**

Immunohistochemical staining of OLFM4 in OSG under normal oesophagus taken from the proximal part of an oesophagectomy specimen resected for Siewert type III junctional tumour in a patient with no BO. Red dashed area and arrow indicates OSG, black arrow indicates OSG duct. Scale bars are 300 μ m and 20 μ m in enlarged image. **(e)** Immunohistochemistry in OSGs from endoscopic biopsy of normal squamous oesophagus in patients with BO. Scale bars are 300 μ m and 50 μ m in enlarged image.

Extended Data Figure 5. Columnar gene-expressing cell detection in patients with and without Barrett's oesophagus

(a) Sankey diagram showing how oesophageal cells from patients with BO (A, B, D; all the cells in the oesophageal biopsies from patient C failed quality control and were not suitable for analysis) and without (E and F) contribute to SC3 clusters generated using oesophageal

cells only. **(b)** Jitter plots showing the expression of columnar (*TFF3* and *KRT7*) and squamous (*KRT14* and *TP63*) genes in cells from **a** coloured by patient (log10tpm) across each cluster. **(c)** Scatter plots showing the expression of *KRT7*, *KRT14* and *TP63* in the two clusters which contain the most *TFF3*-expressing cells (units are log10tpm). **(d)** Immunohistochemical staining of TFF3 in OSGs from normal oesophageal samples obtained endoscopically from two Barrett's patients (scale bars 400µm), with enlarged images (scale bars 50µm). **(e)** Immunohistochemical staining of TFF3 in OSGs under normal oesophagus taken from the proximal part of an oesophagectomy specimen resected for Siewert type III junctional tumour in a patient with no BO. Scale bars are 1000µm and 100µm in enlarged images.

Extended Data Figure 1. Expression of selected tissue and cell defining genes in columnar cells

Boxplots showing the expression of a selection of genes used to define gastric or intestinal tissue cell types (first row of plots are gastric genes, second and third rows are intestinal genes), and a panel of mucin and trefoil factor genes (fourth and fifth rows of plots). Cells included are from all duodenum (n=207), gastric (n=221) and Barrett's samples (n=371), columnar type oesophageal cells are also included as 'OSGs' (n=120, as in Figure 3). Total n=919.

Extended Data Figure 2. Expression of *LEFTY1*, *KRT7* and *KRT20* in BO, duodenal and gastric cells

(a) Jitter plots showing the expression of *LEFTY1*, *KRT7* and *KRT20* in Barrett's, duodenum and gastric cells. **(b)** Immunohistochemical staining of *LEFTY1* in normal tissues and Barrett's oesophagus as indicated. Scale bar 200µm.

Extended Data Figure 11. Sample analysis and quality control

(a) Bar plot showing total number of cells sequenced of each tissue type by patient (A-D and G-H are patients with BO, E-F are patients without BO). Cells passing threshold for inclusion determined by controls (see **Methods**) are in solid colour, excluded cells are shown in faded colour. (b) Violin plots showing number of mapped reads by each tissue type, coloured as in a, in all cells which were sequenced. (c) Plot of number of genes detected (at least one read per gene) against total mapped reads for every sequenced cell and control. (d) Scatter plot showing the correlation between bulk tissue RNA-seq gene counts and an ensemble of single cell gene counts, taken from Barrett's samples from patient A, with Pearson correlation coefficient displayed.

Extended Data Figure 8. StemID applied to human duodenum, Barrett's oesophagus, oesophageal submucosal gland and gastric cells with expression of stem and associated markers in BO.

(a) StemID scores across all clusters computed for duodenum (purple), gastric (green), Barrett's oesophagus (orange) and oesophageal submucosal gland cells (pink) in bar plots from left to right. Scores are calculated from multiplication of the entropy (spread from the

cluster mean) and the number of cluster links arising from a given cluster. Oesophageal submucosal gland cells were deduced by removing squamous (*KRT14+*) cells from all normal oesophagus cells from patients A-D (cells from patients E and F were removed to avoid comparing cells between batches). **(b)** Volcano plots showing significantly expressed genes in the highest scoring cluster corresponding the plot directly above in **a**, from left to right these are duodenum, gastric, Barrett's oesophagus and oesophageal submucosal gland cells. **(c)** Number of Barrett's and OSG cells isolated from each patient and represented in the highest scoring StemID cluster in **Figure 5** (cluster 3). **(d)** Immunohistochemical staining of OLFM4 in control tissues and structures as indicated. Scale bars are 50µm. **(e)** Single cell RNA-seq data showing percentage of cells expressing *OLFM4*, *LEFTY1*, *MUC2* or *CHGA* alone and in all combinations (thresholds for calling a gene 'expressed' were set at the tenth centile to include 90% of cells in which at least one transcript was detected from each gene).

Extended Data Figure 7. Cells from gastric cardia and pylorus have similar RNA compositions.

(a) t-SNE plot of gastric cardia cells (n=109), gastric pylorus cells (n=29), BO cells (n=32), oesophagus cells (n=24) and brain RNA control (n=47) showing transcriptional relationships

between cells obtained from an additional two patients with BO. **(b)** Jitter plots showing expression of selected epithelial genes in gastric cardia, gastric pylorus, BO and oesophagus cells as in **a**.

Figure 4. SPINK4 and ITLN1 mark early goblet cells

(a) Volcano plot showing fold change and p value of genes differentially expressed in the ‘goblet-type’ cell cluster as compared to all other cell clusters (see **Figure 3**). Points coloured

red indicate genes significant at 5% permutation test. Selected highly significant genes are labelled. **(b)** Bar chart showing the percentage of cells in the 'goblet-type' cell cluster (n=98) expressing *MUC2*, *ITLN1* or *SPINK4* alone or in different combinations (thresholds set at the tenth centile to include 90% of cells in which at least one transcript was detected from each gene). **(c)** Triple immunofluorescence staining images of MUC2 (red), ITLN1 (white) and SPINK4 (green) in normal colon from a resection specimen (blue stain is DAPI). Scale bar 100µm. **(d)** Triple immunofluorescence staining images of MUC2 (red), ITLN1 (white) and SPINK4 (green) in normal oesophageal epithelium obtained by endoscopic biopsy (blue stain is DAPI). OSGs encroaching on the surface epithelium are shown in the enlarged images on the right. Scale bars are 200µm and 50µm in enlarged images. **(e)** Triple immunofluorescence staining images of KRT14 (white), KRT7 (green) and MUC2 (red) in an OSG beneath normal squamous epithelium from an endoscopic biopsy of normal squamous epithelium from a patient with BO biopsy (blue stain is DAPI). Scale bar 50µm. **(f)** Representative immunofluorescence staining of Barrett's EMR specimen containing intestinal metaplasia but no dysplasia for MUC2 (red), ITLN1 (white) and SPINK4 (green); nuclei (DAPI) in blue. Scale bars are 400µm and 100µm in enlarged images.

Reviewer #3 (Remarks to the Author):

Barrett's oesophagus is both an important clinical problem and a fascinating phenomenon. Although the nature of the epithelial changes has been hotly debated, no clear concept has emerged. Against this background Owen and co-workers present an scRNAseq analysis that provides compelling evidence to suggest that Barrett's oesophagus may arise from the cells of the oesophageal gland. I am not an expert in scRNAseq analysis, but I believe the work has been carried out to a high standard. The validation experiments, involving mapping expression of LEFTY and other hits, are very strong. The paper is very clearly written.

For me, this work represents a beautiful example of the power of scRNAseq to overturn long-held opinions about tissue function and its link to disease. The findings may ruffle a few feathers but I think the paper will attract widespread interest.

Reviewers' comments:

Reviewer #1 (Remarks to the Author):

The authors addressed many of the issues raised against the paper. This includes the important issue of the selection method that they used using for instance EpCam stainings and improving the images.

The authors also added important extra work the paper and increased the number of patients that were analyzed. This highly improved the paper.

I have a few minor points that should be addressed:

On page 4 the authors state:

In contrast to p63-/KRT7+ RECs, a recent study identified p63+/KRT5+/KRT7+ cells as the cells of origin of BO...'

In this statement they refer to the recent paper of Jiang et al published in Nature. In this paper these cells only were able to differentiate into Barrett like cells in a CDX2 transgenic model. Thus, only on ectopic expression of CDX2, the p63+/KRT5+/KRT7+ cells were able to develop into Barrett resembling metaplasia.

The authors should rephrase the sentence and more precisely state this fact

Many thanks for the clarification – we have updated this sentence as:

'In contrast to p63-/KRT7+ RECs, p63+/KRT5+/KRT7+ cells from the squamocolumnar junction have been proposed as the origin of BO as they have been shown to expand into BO-like cells in a transgenic mouse model which ectopically expresses CDX2 in KRT5+ epithelium.'

On page 5 the authors state:

'This revealed a cell population in BO that expresses the developmental gene (*LEFTY1*) and is distinct from intestinal or gastric cells, but had a highly similar RNA composition to columnar gene expressing cells from normal oesophagus.'

This statement is confusing, since in general the audience is not aware of columnar genes in the normal esophagus. The authors refer to the submucosal glands of the esophagus and therefore should rephrase this sentence.

We have rephrased this sentence as:

'This revealed a cell population in BO that expresses the developmental gene (*LEFTY1*) and is distinct from intestinal or gastric cells, but had a highly similar RNA composition to columnar gene expressing cells from oesophageal submucosal glands in normal oesophagus.'

On page 15 the authors state:

'Notably, *LEFTY1* is regulated by TGF- β signalling⁴² and TGF- β is often ³²³ perturbed in BO.'

The authors should also notify that BMPs regulate *LEFTY1* expression (Smith et al plos genetics 2011). Compared to TGF-beta, BMPs play a major role in development of a Barrett like phenotype (e.g., Mari et al Cell reports 2014).

Thank you for the references, these have been included and this sentence updated to include some detail of the role of BMPs in Barrett's:

'Notably, LEFTY1 is regulated by TGF- β signalling and bone morphogenic proteins (BMPs). Since TGF- β is often perturbed in BO, and BMPs have been shown to play a major role in the development of a BO-like phenotype, it will be interesting to explore these relationships further.'

Reviewer #4 (Remarks to the Author):

In the revised version of the manuscript the authors have significantly extended their analysis and sufficiently addressed all previous concerns of this reviewer.

However, I feel that two things still need to be added to the manuscript. First, the authors should comment on the meaning of the cell grouping entropy shown in Figure 3d. I assume this is a quantity derived by the SC3 method. This should be mentioned in the manuscript and the meaning of this quantity should be briefly explained.

Our measure of entropy is separate to SC3. Measuring entropy provides a means to measure the relative stability of any given grouping of data (see Novoselova et al. Entropy-based cluster validation and estimation of the number of clusters in gene expression data. J Bioinform Comput Biol. 2012).

In Figure 3d, our aim was to show if we manually select our 'clusters', Barrett's oesophagus (BO) and oesophageal submucosal glands (OSGs) together made the most stable grouping, even when other specialised cell types which could drive the clustering were removed.

To clarify this, we have now changed Figure 3d to use 'cluster stability', which is an output from BEARsc (see ref 29), which is the same method we have used elsewhere in the manuscript. We have clarified the above points in the manuscript.

Moreover, the authors should more explicitly show the similarity of BO and OSG cells, by performing a differential gene expression analysis between these two types after exclusion of goblet and enteroendocrine cells (i.e. compare clusters B2/B3 to O3/O4). The differentially expressed genes should be shown in a heatmap and a pathway, GO, or GSEA analysis of these genes should be added to elucidate major differences between these cell types, if existing. This analysis explores the changes that OSG cells undergo upon giving rise to BO cells and could perhaps give a better mechanistic idea of this process.

Differential gene expression is particularly useful at looking for differences between samples. However any signal observed in comparing samples which share profound similarities is less informative. In the early stages of our data analysis, we performed the analysis suggested by this reviewer (please see Figure 1 below). In particular, Figure 1a shows that there is little clustering effect when BO cells (excluding the detectable *CHGA*⁺ enteroendocrine and *MUC2*⁺ goblet specialised cells) and *TFF3*⁺ oesophageal cells (OSGs) are analysed. The annotation bar to the left of Figure 1b shows the hierarchical clustering of cells, revealing that there is no clear BO or OSG clustering pattern. Figure 1c shows the ten most significantly upregulated genes when comparing BO

and OSG cells (following the selection described above). Figure 1d shows the most dysregulated biological processes using the full results of Figure 3c as input. These illustrate that the RNA composition in BO and OSG cells are relatively homogenous and highly similar. Together these data further support our claim that BO and OSG cells share profound transcriptomic similarities, therefore BO may originate from OSG cells. A reduced heatmap and the gene ontology results have now been added as parts e-f, respectively, to Extended Data Figure 6.

Figure 1: (a) t-SNE plot showing the relationships of cells based on gene expression. Cells included are from Barrett's oesophagus (excluding *CHGA*⁺ enteroendocrine and *MUC2*⁺ goblet cells) and oesophageal cells which expressed *TFF3*. (b) A heatmap showing differentially expressed genes (FDR<0.01) between selected BO cells and OSG cells (as in (a)), with gene and sample hierarchical clustering. (c) A bar plot showing the q values of the top ten upregulated genes in BO versus OSG cells. The vertical dashed black line represents a q value of 1e-04. (d) A bar plot showing the FDR values of biological processes from the full results of the differential gene expression analysis shown in (c).